# Fast recovery from a union of subspaces

**Chinmay Hegde**
Iowa State University

**Piotr Indyk**
MIT

**Ludwig Schmidt**
MIT

## Abstract

We address the problem of recovering a high-dimensional but structured vector from linear observations in a general setting where the vector can come from an arbitrary union of subspaces. This setup includes well-studied problems such as compressive sensing and low-rank matrix recovery. We show how to design more efficient algorithms for the union-of-subspace recovery problem by using *approximate* projections. Instantiating our general framework for the low-rank matrix recovery problem gives the fastest provable running time for an algorithm with optimal sample complexity. Moreover, we give fast approximate projections for 2D histograms, another well-studied low-dimensional model of data. We complement our theoretical results with experiments demonstrating that our framework also leads to improved time and sample complexity empirically.

## 1 Introduction

Over the past decade, exploiting low-dimensional structure in high-dimensional problems has become a highly active area of research in machine learning, signal processing, and statistics. In a nutshell, the general approach is to utilize a low-dimensional model of relevant data in order to achieve better prediction, compression, or estimation compared to a "black box" treatment of the ambient high-dimensional space. For instance, the seminal work on compressive sensing and sparse linear regression has shown how to estimate a sparse, high-dimensional vector from a small number of linear observations that essentially depends only on the small sparsity of the vector, as opposed to its large ambient dimension. Further examples of low-dimensional models are low-rank matrices, group-structured sparsity, and general union-of-subspaces models, all of which have found applications in problems such as matrix completion, principal component analysis, compression, and clustering.

These low-dimensional models have a common reason for their success: they capture important structure present in real world data with a formal concept that is suitable for a rigorous mathematical analysis. This combination has led to statistical performance improvements in several applications where the ambient high-dimensional space is too large for accurate estimation from a limited number of samples. However, exploiting the low-dimensional structure also comes at a cost: incorporating the structural constraints into the statistical estimation procedure often results in a more challenging algorithmic problems. Given the growing size of modern data sets, even problems that are solvable in polynomial time can quickly become infeasible. This leads to the following important question: Can we design efficient algorithms that combine (near)-optimal statistical efficiency with good computational complexity?

In this paper, we make progress on this question in the context of recovering a low-dimensional vector from noisy linear observations, which is the fundamental problem underlying both low-rank matrix recovery and compressive sensing / sparse linear regression. While there is a wide range of algorithms for these problems, two approaches for incorporating structure tend to be most common: (i) convex relaxations of the low-dimensional constraint such as the $\ell_1$- or the nuclear norm [19], and (ii) iterative methods based on projected gradient descent, e.g., the IHT (Iterative Hard Thresholding) or SVP (Singular Value Projection) algorithms [5, 15]. Since the convex relaxations are often also solved with first order methods (e.g., FISTA or SVT [6]), the low-dimensional constraint enters both

approaches through a structure-specific projection or proximal operator. However, this projection / proximal operator is often computationally expensive and dominates the overall time complexity (e.g., it requires a singular value decomposition for the low-rank matrix recovery problem).

In this work, we show how to reduce the computational bottleneck of the projection step by using *approximate* projections. Instead of solving the structure-specific projection exactly, our framework allows us to employ techniques from approximation algorithms without increasing the sample complexity of the recovery algorithm. While approximate projections have been used in prior work, our framework is the first to yield provable algorithms for general union-of-subspaces models (such as low-rank matrices) that combine *better running time* with *no loss in sample complexity* compared to their counterparts utilizing exact projections. Overall, we make three contributions:

1. We introduce an algorithmic framework for recovering vectors from linear observations given an arbitrary *union-of-subspaces* model. Our framework only requires approximate projections, which leads to recovery algorithms with significantly better time complexity.

2. We instantiate our framework for the well-studied *low-rank matrix recovery problem*, which yields a provable algorithm combining the optimal sample complexity with the best known time complexity for this problem.

3. We also instantiate our framework for the problem of recovering *2D-histograms* (i.e., piecewise constant matrices) from linear observations, which leads to a better empirical sample complexity than the standard approach based on Haar wavelets.

Our algorithmic framework generalizes recent results for structured sparse recovery [12, 13] and shows that approximate projections can be employed in a wider context. We believe that these notions of approximate projections are useful in further constrained estimation settings and have already obtained preliminary results for structured sparse PCA. For conciseness, we focus on the union-of-subspaces recovery problem in this paper.

**Outline of the paper.** In Section 2, we formally introduce the union-of-subspaces recovery problem and state our main results. Section 3 then explains our algorithmic framework in more detail and Section 4 instantiates the framework for low-rank matrix recovery. Section 5 concludes with experimental results. Due to space constraints, we address our results for 2D histograms mainly in Appendix C of the supplementary material.

## 2 Our contributions

We begin by defining our problem of interest. Our goal is to recover an unknown, structured vector $\theta^* \in \mathbb{R}^d$ from linear observations of the form

$$y = X\theta^* + e \,, \tag{1}$$

where the vector $y \in \mathbb{R}^n$ contains the linear observations / measurements, the matrix $X \in \mathbb{R}^{n \times d}$ is the design / measurement matrix, and the vector $e \in \mathbb{R}^n$ is an arbitrary noise vector. The formal goal is to find an estimate $\hat{\theta} \in \mathbb{R}^d$ such that $\|\hat{\theta} - \theta^*\|_2 \leq C \cdot \|e\|_2$, where $C$ is a fixed, universal constant and $\|\cdot\|_2$ is the standard $\ell_2$-norm (for notational simplicity, we omit the subscript on the $\ell_2$-norm in the rest of the paper). The structure we assume is that the vector $\theta^*$ belongs to a *subspace model*:

**Definition 1** (Subspace model). *A subspace model $\mathbb{U}$ is a set of linear subspaces. The set of vectors associated with the subspace model $\mathbb{U}$ is $\mathcal{M}(\mathbb{U}) = \{\theta \,|\, \theta \in U \text{ for some } U \in \mathbb{U}\}$.*

A subspace model is a natural framework generalizing many of the low-dimensional data models mentioned above. For example, the set of sparse vectors with $s$ nonzeros can be represented with $\binom{d}{s}$ subspaces corresponding to the $\binom{d}{s}$ possible sparse support sets. The resulting problem of recovering $\theta^*$ from observations of the form (1) then is the standard compressive sensing / sparse linear regression problem. Structured sparsity is a direct extension of this formulation in which we only include a smaller set of allowed supports, e.g., supports corresponding to group structures.

Our framework also includes the case where the union of subspaces is taken over an infinite set: we can encode the low-rank matrix recovery problem by letting $\mathbb{U}$ be the set of rank-$r$ matrix subspaces, i.e., each subspace is given by a set of $r$ orthogonal rank-one matrices. By considering the singular

value decomposition, it is easy to see that every rank-$r$ matrix can be written as the linear combination of $r$ orthogonal rank-one matrices.

Next, we introduce related notation. For a linear subspace $U$ of $\mathbb{R}^d$, let $P_U \in \mathbb{R}^{d \times d}$ be the orthogonal projection onto $U$. We denote the orthogonal complement of the subspace $U$ with $U^\perp$ so that $\theta = P_U \theta + P_{U^\perp} \theta$. We extend the notion of adding subspaces (i.e., $U + V = \{u + v \mid u \in U \text{ and } v \in V\}$) to subspace models: the sum of two subspace models $\mathbb{U}$ and $\mathbb{V}$ is $\mathbb{U} \oplus \mathbb{V} = \{U + V \mid U \in \mathbb{U} \text{ and } V \in \mathbb{V}\}$. We denote the $k$-wise sum of a subspace model with $\oplus^k \mathbb{U} = \mathbb{U} \oplus \mathbb{U} \oplus \ldots \oplus \mathbb{U}$.

Finally, we introduce a variant of the well-known restricted isometry property (RIP) for subspace models. The RIP is a common regularity assumption for the design matrix $X$ that is often used in compressive sensing and low-rank matrix recovery in order to decouple the analysis of algorithms from concrete sampling bounds.[1] Formally, we have:

**Definition 2** (Subspace RIP). *Let $X \in \mathbb{R}^{n \times d}$, let $\mathbb{U}$ be a subspace model, and let $\delta \geq 0$. Then $X$ satisfies the $(\mathbb{U}, \delta)$-subspace RIP if for all $\theta \in \mathcal{M}(\mathbb{U})$ we have $(1 - \delta)\|\theta\|^2 \leq \|X\theta\|^2 \leq (1 + \delta)\|\theta\|^2$.*

## 2.1 A framework for recovery algorithms with approximate projections

Considering the problem (1) and the goal of estimating under the $\ell_2$-norm, a natural algorithm is projected gradient descent with the constraint set $\mathcal{M}(\mathbb{U})$. This corresponds to iterations of the form

$$\hat{\theta}^{i+1} \;\leftarrow\; P_{\mathbb{U}}(\hat{\theta}^i - \eta \cdot X^T(X\hat{\theta}^i - y)) \tag{2}$$

where $\eta \in \mathbb{R}$ is the step size and we have extended our notation so that $P_{\mathbb{U}}$ denotes a projection onto the set $\mathcal{M}(\mathbb{U})$. Hence we require an oracle that projects an arbitrary vector $b \in \mathbb{R}^d$ into a subspace model $\mathbb{U}$, which corresponds to finding a subspace $U \in \mathbb{U}$ so that $\|b - P_U b\|$ is minimized. Recovery algorithms of the form (2) have been proposed for various instances of the union-of-subspaces recovery problem and are known as Iterative Hard Thresholding (IHT) [5], model-IHT [1], and Singular Value Projection (SVP) [15]. Under regularity conditions on the design matrix $X$ such as the RIP, these algorithms find accurate estimates $\hat{\theta}$ from an asymptotically optimal number of samples. However, for structures more complicated than plain sparsity (e.g., group sparsity or a low-rank constraint), the projection oracle is often the computational bottleneck.

To overcome this barrier, we propose two complementary notions of *approximate* subspace projections. Note that for an exact projection, we have that $\|b\|^2 = \|b - P_{\mathbb{U}} b\|^2 + \|P_{\mathbb{U}} b\|^2$. Hence minimizing the "tail" error $\|b - P_{\mathbb{U}} b\|$ is equivalent to maximizing the "head" quantity $\|P_{\mathbb{U}} b\|$. Instead of minimizing / maximizing these quantities exactly, the following definitions allow a *constant factor* approximation:

**Definition 3** (Approximate tail projection). *Let $\mathbb{U}$ and $\mathbb{U}_{\mathcal{T}}$ be subspace models and let $c_{\mathcal{T}} \geq 0$. Then $\mathcal{T} : \mathbb{R}^d \to \mathbb{U}_{\mathcal{T}}$ is a $(c_{\mathcal{T}}, \mathbb{U}, \mathbb{U}_{\mathcal{T}})$-approximate tail projection if the following guarantee holds for all $b \in \mathbb{R}^d$: The returned subspace $U = \mathcal{T}(b)$ satisfies $\|b - P_U b\| \leq c_{\mathcal{T}} \|b - P_{\mathbb{U}} b\|$.*

**Definition 4** (Approximate head projection). *Let $\mathbb{U}$ and $\mathbb{U}_{\mathcal{H}}$ be subspace models and let $c_{\mathcal{H}} > 0$. Then $\mathcal{H} : \mathbb{R}^d \to \mathbb{U}_{\mathcal{H}}$ is a $(c_{\mathcal{H}}, \mathbb{U}, \mathbb{U}_{\mathcal{H}})$-approximate head projection if the following guarantee holds for all $b \in \mathbb{R}^d$: The returned subspace $U = \mathcal{H}(b)$ satisfies $\|P_U b\| \geq c_{\mathcal{H}} \|P_{\mathbb{U}} b\|$.*

It is important to note that the two definitions are distinct in the sense that a constant-factor head approximation does not imply a constant-factor tail approximation, or vice versa (to see this, consider a vector with a very large or very small tail error, respectively). Another feature of these definitions is that the approximate projections are allowed to choose subspaces from a potentially larger subspace model, i.e., we can have $\mathbb{U} \subsetneq \mathbb{U}_{\mathcal{H}}$ (or $\mathbb{U}_{\mathcal{T}}$). This is a useful property when designing approximate head and tail projection algorithms as it allows for *bicriterion* approximation guarantees.

We now state the main result for our new recovery algorithm. In a nutshell, we show that using *both* notions of approximate projections achieves the same statistical efficiency as using exact projections. As we will see in later sections, the weaker approximate projection guarantees allow us to design algorithms with a significantly better time complexity than their exact counterparts. To simplify the following statement, we defer the precise trade-off between the approximation ratios to Section 3.

**Theorem 5** (informal). *Let $\mathcal{H}$ and $\mathcal{T}$ be approximate head and tail projections with constant approximation ratios, and let the matrix $X$ satisfy the $(\oplus^c \mathbb{U}, \delta)$-subspace RIP for a sufficiently large constant $c$ and a sufficiently small constant $\delta$. Then there is an algorithm* AS-IHT *that returns an estimate $\hat{\theta}$ such that $\|\hat{\theta} - \theta^*\| \leq C\|e\|$. The algorithm requires $O(\log\|\theta\|/\|e\|)$ multiplications with $X$ and $X^T$, and $O(\log\|\theta\|/\|e\|)$ invocations of $\mathcal{H}$ and $\mathcal{T}$.*

Up to constant factors, the requirements on the RIP of $X$ in Theorem 5 are the same as for exact projections. As a result, our sample complexity is only affected by a constant factor through the use of approximate projections, and our experiments in Section 5 show that the empirical loss in sample complexity is negligible. Similarly, the number of iterations $O(\log\|\theta\|/\|e\|)$ is also only affected by a constant factor compared to the use of exact projections [5, 15]. Finally, it is worth mentioning that using two notions of approximate projections is crucial: prior work in the special case of structured sparsity has already shown that only one type of approximate projection is not sufficient for strong recovery guarantees [13].

## 2.2 Low-rank matrix recovery

We now instantiate our new algorithmic framework for the low-rank matrix recovery problem. Variants of this problem are widely studied in machine learning, signal processing, and statistics, and are known under different names such as matrix completion, matrix sensing, and matrix regression. As mentioned above, we can incorporate the low-rank matrix structure into our general union-of-subspaces model by considering the union of all low-rank matrix subspaces. For simplicity, we state the following bounds for the case of square matrices, but all our results also apply to rectangular matrices. Formally, we assume that $\theta^* \in \mathbb{R}^d$ is the vectorized form of a rank-$r$ matrix $\Theta^* \in \mathbb{R}^{d_1 \times d_1}$ where $d = d_1^2$ and typically $r \ll d_1$. Seminal results have shown that it is possible to achieve the subspace-RIP for low-rank matrices with only $n = O(r \cdot d_1)$ linear observations, which can be much smaller than the total dimensionality of the matrix $d_1^2$. However, the bottleneck in recovery algorithms is often the singular value decomposition (SVD), which is necessary for both exact projections and soft thresholding operators and has a time complexity of $O(d_1^3)$.

Our new algorithmic framework for approximate projections allows us to leverage recent results on *approximate* SVDs. We show that it is possible to compute both head and tail projections for low-rank matrices in $\widetilde{O}(r \cdot d_1^2)$ time, which is significantly faster than the $O(d_1^3)$ time for an exact SVD in the relevant regime where $r \ll d_1$. Overall, we get the following result.

**Theorem 6.** *Let $X \in \mathbb{R}^{n \times d}$ be a matrix with subspace-RIP for low-rank matrices, and let $T_X$ denote the time to multiply a $d$-dimensional vector with $X$ or $X^T$. Then there is an algorithm that recovers an estimate $\hat{\theta}$ such that $\|\hat{\theta} - \theta^*\| \leq C\|e\|$. Moreover, the algorithm runs in time $\widetilde{O}(T_X + r \cdot d_1^2)$.*

In the regime where multiplication with the matrix $X$ is fast, the time complexity of the projection dominates the time complexity of the recovery algorithms. For instance, structured observations such as a subsampled Fourier matrix achieve $T_X = \widetilde{O}(d_1^2)$; see Appendix D for details. Here, our algorithm runs in time $\widetilde{O}(r \cdot d_1^2)$, which is the first provable running time faster than the $O(d_1^3)$ bottleneck given by a single exact SVD. While prior work has suggested the use of approximate SVDs in low-rank matrix recovery [9], our results are the first that give a provably better time complexity for this combination of projected gradient descent and approximate SVDs. Hence Theorem 6 can be seen as a theoretical justification for the heuristic use of approximate SVDs.

Finally, we remark that Theorem 6 does not directly cover the low-rank matrix completion case because the subsampling operator does not satisfy the low-rank RIP [9]. To clarify our use of approximate SVDs, we focus on the RIP setting in our proofs, similar to recent work on low-rank matrix recovery [7, 22]. We believe that similar results as for SVP [15] also hold for our algorithm, and our experiments in Section 5 show that our algorithm works well for low-rank matrix completion.

## 2.3 2D-histogram recovery

Next, we instantiate our new framework for 2D-histograms, another natural low-dimensional model. As before, we think of the vector $\theta^* \in \mathbb{R}^d$ as a matrix $\Theta \in \mathbb{R}^{d_1 \times d_1}$ and assume the square case for simplicity (again, our results also apply to rectangular matrices). We say that $\Theta$ is a $k$-histogram if the coefficients of $\Theta$ can be described as $k$ axis-aligned rectangles on which $\Theta$ is constant. This definition

is a generalization of 1D-histograms to the two-dimensional setting and has found applications in several areas such as databases and density estimation. Moreover, the theoretical computer science community has studied sketching and streaming algorithms for histograms, which is essentially the problem of recovering a histogram from linear observations. While the wavelet tree model with Haar wavelets give the correct sample complexity of $n = O(k \log d)$ for 1D-histograms, the wavelet tree approach incurs a *suboptimal* sample complexity of $O(k \log^2 d)$ for 2D-histograms. It is possible to achieve the optimal sample complexity $O(k \log d)$ also for 2D-histograms, but the corresponding exact projection requires a complicated dynamic program (DP) with time complexity $O(d_1^5 k^2)$, which is impractical for all but very small problem dimensions [18].

We design significantly faster *approximate* projection algorithms for 2D histograms. Our approach is based on an approximate DP [18] that we combine with a Lagrangian relaxation of the $k$-rectangle constraint. Both algorithms have parameters for controlling the trade-off between the size of the output histogram, the approximation ratio, and the running time. As mentioned above, the bicriterion nature of our approximate head and tail guarantees becomes useful here. In the following two theorems, we let $\mathbb{U}_k$ be the subspace model of 2D histograms consisting of $k$-rectangles.

**Theorem 7.** *Let $\zeta > 0$ and $\varepsilon > 0$ be arbitrary. Then there is an $(1 + \varepsilon, \mathbb{U}_k, \mathbb{U}_{c \cdot k})$-approximate tail projection for 2D histograms where $c = O(1/\zeta^2 \varepsilon)$. Moreover, the algorithm runs in time $\widetilde{O}(d^{1+\zeta})$.*

**Theorem 8.** *Let $\zeta > 0$ and $\varepsilon > 0$ be arbitrary. Then there is an $(1 - \varepsilon, \mathbb{U}_k, \mathbb{U}_{c \cdot k})$-approximate head projection for 2D histograms where $c = O(1/\zeta^2 \varepsilon)$. Moreover, the algorithm runs in time $\widetilde{O}(d^{1+\zeta})$.*

Note that both algorithms offer a running time that is *almost linear*, and the small polynomial gap to a linear running time can be controlled as a trade-off between computational and statistical efficiency (a larger output histogram requires more samples to recover). While we provide rigorous proofs for the approximation algorithms as stated above, we remark that we do not establish an overall recovery result similar to Theorem 6. The reason is that the approximate head projection is competitive with respect to $k$-histograms, but not with the space $\mathbb{U}_k \oplus \mathbb{U}_k$, i.e., the sum of two $k$-histogram subspaces. The details are somewhat technical and we give a more detailed discussion in Appendix C.3. However, under a natural structural conjecture about sums of $k$-histogram subspaces, we obtain a similar result as Theorem 6. Moreover, we experimentally demonstrate that the sample complexity of our algorithms already improves over wavelets for $k$-histograms of size $32 \times 32$.

Finally, we note that our DP approach also generalizes to $\gamma$-dimensional histograms for any constant $\gamma \geq 2$. As the dimension of the histogram structure increases, the gap in sample complexity between our algorithm and the prior wavelet-based approach becomes increasingly wide and scales as $O(k\gamma \log d)$ vs $O(k \log^\gamma d)$. For simplicity, we limit our attention to the 2D case described above.

## 2.4 Related work

Recently, there have been several results on approximate projections in the context of recovering low-dimensional structured vectors. (see [12, 13] for an overview). While these approaches also work with approximate projections, they only apply to less general models such as dictionary sparsity [12] or structured sparsity [13] and do not extend to the low-rank matrix recovery problem we address. Among recovery frameworks for general union-of-subspaces models, the work closest to ours is [4], which also gives a generalization of the IHT algorithm. It is important to note that [4] addresses approximate projections, but requires *additive error* approximation guarantees instead of the weaker *relative error* approximation guarantees required by our framework. Similar to the structured sparsity case in [13], we are not aware of any algorithms for low-rank or histogram projections that offer additive error guarantees faster than an exact projection. Overall, our recovery framework can be seen as a generalization of the approaches in [13] and [4].

Low-rank recovery has received a tremendous amount of attention over the past few years, so we refer the reader to the recent survey [9] for an overview. When referring to prior work on low-rank recovery, it is important to note that the fastest known running time for an exact low-rank SVD (even for rank 1) of a $d_1 \times d_2$ matrix is $O(d_1 d_2 \min(d_1, d_2))$. Several papers provide rigorous proofs for low-rank recovery using exact SVDs and then refer to Lanczos methods such as PROPACK [16] while accounting a time complexity of $O(d_1 d_2 r)$ for a rank-$r$ SVD. While Lanczos methods can be faster than exact SVDs in the presence of singular value gaps, it is important to note that all rigorous results for Lanczos SVDs either have a polynomial dependence on the approximation ratio or singular

value gaps [17, 20]. No prior work on low-rank recovery establishes such singular value gaps for the inputs to the SVD subroutines (and such gaps would be necessary for *all* iterates in the recovery algorithm). In contrast, we utilize recent work on gap-independent approximate SVDs [17], which enables us to give rigorous guarantees for the entire recovery algorithm. Our results can be seen as justification for the heuristic use of Lanczos methods in prior work.

The paper [2] contains an analysis of an approximate SVD in combination with an iterative recovery algorithm. However, [2] only uses an approximate tail projection, and as a result the approximation ratio $c_{\mathcal{T}}$ must be very close to 1 in order to achieve a good sample complexity. Overall, this leads to a time complexity that does not provide an asymptotic improvement over using exact SVDs.

Recently, several papers have analyzed a non-convex approach to low-rank matrix recovery via factorized gradient descent [3, 7, 22–24]. While these algorithms avoid SVDs in the iterations of the gradient method, the overall recovery proofs still require an exact SVD in the initialization step. In order to match the sample complexity of our algorithm or SVP, the factorized gradient methods require multiple SVDs for this initialization [7, 22]. As a result, our algorithm offers a better provable time complexity. We remark that [7, 22] use SVP for their initialization, so combining our faster version of SVP with factorized gradient descent might give the best overall performance.

As mentioned earlier, 1D and 2D histograms have been studied extensively in several areas such as databases [8, 14] and density estimation. They are typically used to summarize "count vectors", with each coordinate of the vector $\theta$ corresponding the number of items with a given value in some data set. Computing linear sketches of such vectors, as well as efficient methods for recovering histogram approximations from those sketches, became key tools for designing space efficient dynamic streaming algorithms [10, 11, 21]. For 1D histograms it is known how to achieve the optimal sketch length bound of $n = O(k \log d)$: it can be obtained by representing $k$-histograms using a tree of $O(k \log d)$ wavelet coefficients as in [10] and then using the structured sparse recovery algorithm of [1]. However, applying this approach to 2D histograms leads to a sub-optimal bound of $O(k \log^2 d)$.

## 3 An algorithm for recovery with approximate projections

We now introduce our algorithm for recovery from general subspace models using only approximate projections. The pseudo code is formally stated in Algorithm 1 and can be seen as a generalization of IHT [5]. Similar to IHT, we give a version without step size parameter here in order to simplify the presentation (it is easy to introduce a step size parameter in order to fine-tune constant factors). To clarify the connection with projected gradient descent as stated in Equation (2), we use $\mathcal{H}(b)$ (or $\mathcal{T}(b)$) as a function from $\mathbb{R}^d$ to $\mathbb{R}^d$ here. This function is then understood to be $b \mapsto P_{\mathcal{H}(b)}b$, i.e., the orthogonal projection of $b$ onto the subspace identified by $\mathcal{H}(b)$.

---
**Algorithm 1** Approximate Subspace-IHT

1: **function** AS-IHT$(y, X, t)$
2: $\quad \hat{\theta}^0 \leftarrow 0$
3: $\quad$ **for** $i \leftarrow 0, \ldots, t$ **do**
4: $\quad\quad b^i \leftarrow X^T(y - X\hat{\theta}^i)$
5: $\quad\quad \hat{\theta}^{i+1} \leftarrow \mathcal{T}(\hat{\theta}^i + \mathcal{H}(b^i))$
6: $\quad$ **return** $\hat{\theta} \leftarrow \hat{\theta}^{t+1}$

---

The main difference to "standard" projected gradient descent is that we apply a projection to *both* the gradient step and the new iterate. Intuitively, the head projection ensures two points: (i) The result of the head projection on $b^i$ still contains a constant fraction of the residual $\theta^* - \hat{\theta}^i$ (see Lemma 13 in Appendix A). (ii) The input to the tail approximation is close enough to the constraint set $\mathbb{U}$ so that the tail approximation does not prevent the overall convergence. In a nutshell, the head projection "denoises" the gradient so that we can then safely apply an approximate tail projection (as pointed out in [13], only applying an approximate tail projection fails precisely because of "noisy" updates). Formally, we obtain the following theorem for each iteration of AS-IHT (see Appendix A.1 for the corresponding proof):

**Theorem 9.** *Let $\hat{\theta}^i$ be the estimate computed by* AS-IHT *in iteration $i$ and let $r^{i+1} = \theta^* - \hat{\theta}^{i+1}$ be the corresponding residual. Moreover, let $\mathbb{U}$ be an arbitrary subspace model. We also assume:*

- *$y = X\theta^* + e$ as in Equation (1) with $\theta^* \in \mathcal{M}(\mathbb{U})$.*
- *$\mathcal{T}$ is a $(c_\mathcal{T}, \mathbb{U}, \mathbb{U}_\mathcal{T})$-approximate tail projection.*
- *$\mathcal{H}$ is a $(c_\mathcal{H}, \mathbb{U} \oplus \mathbb{U}_\mathcal{T}, \mathbb{U}_\mathcal{H})$-approximate head projection.*
- *The matrix $X$ satisfies the $(\mathbb{U} \oplus \mathbb{U}_\mathcal{T} \oplus \mathbb{U}_\mathcal{H}, \delta)$-subspace RIP.*

*Then the residual error of the next iterate, i.e., $r^{i+1} = \theta^* - \hat{\theta}^{i+1}$ satisfies*

$$\left\| r^{i+1} \right\| \leq \eta \left\| r^i \right\| + \rho \|e\| \,,$$

*where* $\qquad \eta = (1 + c_\mathcal{T})\left(\delta + \sqrt{1 - \eta_0^2}\right) \,, \qquad\qquad \rho = (1 + c_\mathcal{T})\left(\dfrac{\eta_0 \rho_0}{\sqrt{1 - \eta_0^2}} + \sqrt{1 + \delta}\right) \,,$

$\qquad\qquad \eta_0 = c_\mathcal{H}(1 - \delta) - \delta \,, \qquad\qquad$ *and* $\qquad \rho_0 = (1 + c_\mathcal{H})\sqrt{1 + \delta} \,.$

The important conclusion of Theorem 9 is that AS-IHT still achieves linear convergence when the approximation ratios $c_\mathcal{T}, c_\mathcal{H}$ are sufficiently close to 1 and the RIP-constant $\delta$ is sufficiently small. For instance, our approximation algorithms for both low-rank matrices offer such approximation guarantees. We can also achieve a sufficiently small value of $\delta$ by using a larger number of linear observations in order to strengthen the RIP guarantee (see Appendix D). Hence the use of approximate projections only affects the theoretical sample complexity bounds by constant factors. Moreover, our experiments show that approximate projections achieve essentially the same empirical sample complexity as exact projections (see Section 5).

Given sufficiently small / large constants $c_\mathcal{T}$, $c_\mathcal{H}$, and $\delta$, it is easy to see that the linear convergence implied by Theorem 9 directly gives the recovery guarantee and bound on the number of iterations stated in Theorem 5 (see Appendix A.1). However, in some cases it might not be possible to design approximation algorithms with constants $c_\mathcal{T}$ and $c_\mathcal{H}$ sufficiently close to 1 (in contrast, increasing the sample complexity by a constant factor in order to improve $\delta$ is usually a direct consequence of the RIP guarantee or similar statistical regularity assumptions). In order to address this issue, we show how to "boost" an approximate head projection so that the new approximation ratio is arbitrarily close to 1. While this also increases the size of the resulting subspace model, this increase usually affects the sample complexity only by constant factors as before. Note that for any fixed $c_\mathcal{T}$, setting $c_\mathcal{H}$ sufficiently close to 1 and $\delta$ sufficiently small leads to a convergence rate $\eta < 1$ (c.f. Theorem 9). Hence head boosting enables a linear convergence result for *any* initial combinations of $c_\mathcal{T}$ and $c_\mathcal{H}$ while only increasing the sample complexity by a constant factor (see Appendix A.3). Formally, we have the following theorem for head boosting, the proof of which we defer to Appendix A.2.

**Theorem 10.** *Let $\mathcal{H}$ be a $(c_\mathcal{H}, \mathbb{U}, \mathbb{U}_\mathcal{H})$-approximate head projection running in time $O(T)$, and let $\varepsilon > 0$. Then there is a constant $c = c_{\varepsilon, c_\mathcal{H}}$ that depends only on $\varepsilon$ and $c_\mathcal{H}$ such that we can construct a $(1 - \varepsilon, \mathbb{U}, \oplus^c \mathbb{U}_\mathcal{H})$-approximate head projection running in time $O(c(T + T_1' + T_2'))$ where $T_1'$ is the time needed to apply a projection onto a subspace in $\oplus^c \mathbb{U}_\mathcal{H}$, and $T_2'$ is the time needed to find an orthogonal projector for the sum of two subspaces in $\oplus^c \mathbb{U}_\mathcal{H}$.*

We note that the idea of head boosting has already appeared in the context of structured sparse recovery [13]. However, the proof of Theorem 10 is more involved because the subspace in a general subspace model can have arbitrary angles (for structured sparsity, the subspaces are either parallel or orthogonal in each coordinate).

## 4   Low-rank matrix recovery

We now instantiate our framework for recovery from a subspace model to the low-rank matrix recovery problem. Since we already have proposed the top-level recovery algorithm in the previous section, we only have to provide the problem-specific head and tail approximation algorithms here. We use the following result from prior work on approximate SVDs.

**Fact 11** ([17])**.** *There is an algorithm* APPROXSVD *with the following guarantee. Let $A \in \mathbb{R}^{d_1 \times d_2}$ be an arbitrary matrix, let $r \in \mathbb{N}$ be the target rank, and let $\varepsilon > 0$ be the desired accuracy. Then with probability $1 - \psi$,* APPROXSVD$(A, r, \varepsilon)$ *returns an orthonormal set of vectors $z_1, \ldots, z_r \in \mathbb{R}^{d_1}$ such that for all $i \in [r]$, we have*

$$\left| z_i^T A A^T z_i - \sigma_i^2 \right| \leq \varepsilon \sigma_{r+1}^2 \,, \tag{3}$$

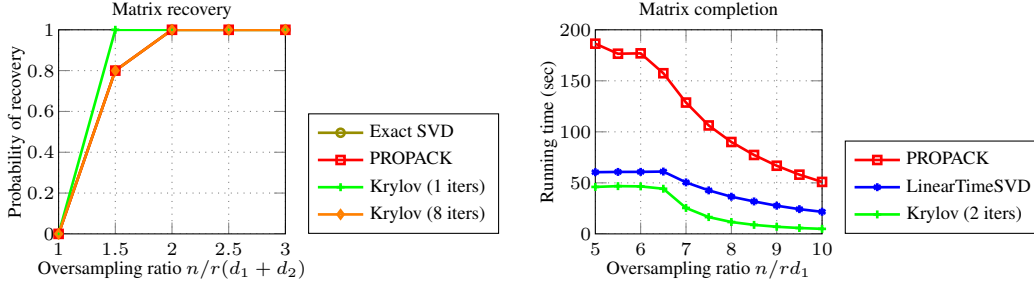

Figure 1: Left: Results for a low-rank matrix recovery experiment using subsampled Fourier measurements. SVP / IHT with one iteration of a block Krylov SVD achieves the same phase transition as SVP with an exact SVD. Right: Results for a low-rank matrix completion problem. SVP / IHT with a block Krylov SVD achieves the best running time and is about $4-8$ times faster than PROPACK.

where $\sigma_i$ is the $i$-th largest singular value of $A$. Furthermore, let $Z \in \mathbb{R}^{d_1 \times r}$ be the matrix with columns $z_i$. Then we also have

$$\left\| A - ZZ^T A \right\|_F \leq (1+\varepsilon)\|A - A_r\|_F ,\qquad(4)$$

where $A_r$ is the best rank-$r$ Frobenius-norm approximation of $A$. Finally, the algorithm runs in time $O\left(\frac{d_1 d_2 r \log(d_2/\psi)}{\sqrt{\varepsilon}} + \frac{d_1 r^2 \log^2(d_2/\psi)}{\varepsilon} + \frac{r^3 \log^3(d_2/\psi)}{\varepsilon^{3/2}}\right)$.

It is important to note that the above results hold for *any* input matrix and do not require singular value gaps. The guarantee (4) directly gives a tail approximation guarantee for the subspace corresponding to the matrix $ZZ^T A$. Moreover, we can convert the guarantee (3) to a head approximation guarantee (see Theorem 18 in Appendix B for details). Since the approximation $\varepsilon$ only enters the running time in the approximate SVD, we can directly combine these approximate projections with Theorem 9, which then yields Theorem 6 (see Appendix B.1 for details).[2] Empirically, we show in the next section that a very small number of iterations in APPROXSVD already suffices for accurate recovery.

## 5   Experiments

We now investigate the empirical performance of our proposed algorithms. We refer the reader to Appendix E for more details about the experiments and results for 2D histograms.

Considering our theoretical results on approximate projections for low-rank recovery, one important empirical question is how the use of approximate SVDs such as [17] affects the sample complexity of low-rank matrix recovery. For this, we perform a standard experiment and use several algorithms to recover an image of the MIT logo from subsampled Fourier measurements (c.f. Appendix D). The MIT logo has also been used in prior work [15, 19]; we use an image with dimensions $200 \times 133$ and rank 6 (see Appendix E). We limit our attention here to variants of SVP because the algorithm has good empirical performance and has been used as baseline in other works on low-rank recovery. Figure 1 shows that SVP / IHT combined with a single iteration of a block Krylov SVD [17] achieves the same phase transition as SVP with exact SVDs. This indicates that the use of approximate projections for low-rank recovery is not only theoretically sound but can also lead to practical algorithms. In Appendix E we also show corresponding running time results demonstrating that the block Krylov SVD also leads to the fastest recovery algorithm.

We also study the performance of approximate SVDs for the matrix completion problem. We generate a symmetric matrix of size $2048 \times 2048$ with rank $r = 50$ and observe a varying number of entries of the matrix. The approximation errors of the various algorithms are again comparable and reported in Appendix E. Figure 1 shows the resulting running times for several sampling ratios. Again, SVP combined with a block Krylov SVD [17] achieves the best running time. Depending on the oversampling ratio, the block Krylov approach (now with two iterations) is 4 to 8 times faster than SVP with PROPACK.

## Footnotes

[1]Note that exact recovery from arbitrary linear observations is already an NP-hard problem in the noiseless case, and hence regularity conditions on the design matrix $X$ are necessary for efficient algorithms. While there are more general regularity conditions such as the restricted eigenvalue property, we state our results here under the RIP assumption in order to simplify the presentation of our algorithmic framework.

[2]We remark that our definitions require head and tail projections to be *deterministic*, while the approximate SVD is *randomized*. However, the running time of APPROXSVD depends only logarithmically on the failure probability, and it is straightforward to apply a union bound over all iterations of AS-IHT. Hence we ignore these details here to simplify the presentation.

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
