[Supplementary Material]

# A  Proofs for our recovery framework using approximate projections

In this appendix, we provide the convergence proof of our recovery algorithm AS-IHT and related results. Before we begin with the analysis of AS-IHT, we first establish useful consequences of the subspace RIP. For convenience, we recall the definition of the subspace RIP:

**Definition 2** (Subspace RIP). *Let $X \in \mathbb{R}^{n \times d}$, let $\mathbb{U}$ be a subspace model, and let $\delta \geq 0$. Then $X$ satisfies the $(\mathbb{U}, \delta)$-subspace RIP if for all $\theta \in \mathcal{M}(\mathbb{U})$ we have $(1-\delta)\|\theta\|^2 \leq \|X\theta\|^2 \leq (1+\delta)\|\theta\|^2$.*

Note that the above definition implies that $\|XP_U\| \leq \sqrt{1+\delta}$ for all $U \in \mathbb{U}$, where $\|XP_U\|$ denotes the spectral norm of $XP_U$. The following lemma summarizes further consequences of the subspace RIP.

**Lemma 12** (Needell, Tropp 2008). *Let $X \in \mathbb{R}^{n \times d}$ be a matrix satisfying the $(\mathbb{U}, \delta)$-subspace RIP. Moreover, let $U \in \mathbb{U}$ be a subspace in the model. Then the following properties hold for all $\theta \in \mathbb{R}^d$ and $y \in \mathbb{R}^n$:*

$$\left\| P_U X^T y \right\| \leq \sqrt{1+\delta}\|y\| \, , \tag{5}$$

$$\left\| P_U X^T X P_U \theta \right\| \leq (1+\delta)\|\theta\| \, , \tag{6}$$

$$\left\| (I - P_U X^T X P_U)\theta \right\| \leq \delta\|\theta\| \, . \tag{7}$$

*Proof.* Equations (5) and (6) follow directly from the bound on the spectral norm of $XP_U$ (which has the same spectral norm as $P_U X^T$).

For Equation (7), consider the eigendecomposition of the symmetric matrix $P_U X^T X P_U$. All eigenvalues are in the interval $[1-\delta, 1+\delta]$. Hence forming $I - P_U X^T X P_U$ shifts all eigenvalues into the interval $[-\delta, \delta]$, which implies the spectral norm bound in Equation (7). □

## A.1  Convergence of AS-IHT

We first prove an important lemma for the convergence proof of our algorithm AS-IHT. In a nutshell, the lemma shows that the approximate head projection captures a significant fraction of the residual vector.

**Lemma 13.** *Let $\mathbb{U}$, $\mathbb{U}_{\mathcal{T}}$, and $\mathbb{U}_{\mathcal{H}}$ be subspace models, and let $\theta^* \in \mathbb{U}$ and $\hat{\theta}^i \in \mathbb{U}_{\mathcal{T}}$ be vectors such that $y = X\theta^* + e$ as in Equation (1) and $\hat{\theta}^i$ is arbitrary. We also assume that the matrix $X$ satisfies the $(\mathbb{U} \oplus \mathbb{U}_{\mathcal{T}} \oplus \mathbb{U}_{\mathcal{H}}, \delta)$-subspace RIP. Furthermore, let $\mathcal{H}$ be a $(c_{\mathcal{H}}, \mathbb{U} \oplus \mathbb{U}_{\mathcal{T}}, \mathbb{U}_{\mathcal{H}})$-approximate head projection. Finally, we define the residual $r^i = \theta^* - \hat{\theta}^i$, the update vector $b^i = X^T(y - X\hat{\theta}^i) = X^T X r^i + X^T e$, and the subspace $U = \mathcal{H}(b^i)$. Then we have*

$$\left\| P_{U^\perp} r^i \right\| \leq \sqrt{1 - \eta_0^2}\left\| r^i \right\| + \frac{\eta_0 \rho_0}{\sqrt{1 - \eta_0^2}}\|e\| \, , \tag{8}$$

*where*

$$\eta_0 = c_{\mathcal{H}}(1-\delta) - \delta \qquad and \qquad \rho_0 = (1 + c_{\mathcal{H}})\sqrt{1+\delta} \, .$$

*We assume that $c_{\mathcal{H}}$ and $\delta$ are such that $\eta_0 < 1$.*

*Proof.* We first give a lower bound on the part of the residual $\|P_U r\|$ that is "captured" by the approximate head projection. We establish this lower bound via the norm of the update vector $\left\| P_U b^i \right\|$. Let $V \in \mathbb{U} \oplus \mathbb{U}_{\mathcal{T}}$ be a subspace such that $r^i \in V$ (note that this is always possible because $\theta \in \mathcal{M}(\mathbb{U})$ and $\hat{\theta}^i \in \mathcal{M}(\mathbb{U}_{\mathcal{T}})$). Using the approximate head projection property of $\mathcal{H}$, we get

$$\begin{aligned} \left\| P_U b^i \right\| &\geq c_{\mathcal{H}}\left\| P_V b^i \right\| \\ &= c_{\mathcal{H}}\left\| P_V X^T X r^i + P_V X^T e \right\| \\ &\geq c_{\mathcal{H}}\left\| P_V X^T X P_V r^i \right\| - c_{\mathcal{H}}\left\| P_V X^T e \right\| \tag{9} \\ &\geq c_{\mathcal{H}}(1-\delta)\left\| r^i \right\| - c_{\mathcal{H}}\sqrt{1+\delta}\|e\| \, . \tag{10} \end{aligned}$$

Equation (9) follows from the triangle inequality and the definition of $V$, which implies $P_V r^i = r^i$. Equation (10) uses Lemma 12 twice.

We now establish an upper bound on $\|P_U b^i\|$:

$$
\begin{aligned}
\|P_U b^i\| &= \|P_U X^T X r^i + P_U X^T e\| \\
&= \|P_U X^T X r^i - P_U r^i + P_U r^i + P_U X^T e\| \\
&\leq \|P_U (X^T X r^i - r^i)\| + \|P_U r^i\| + \|P_U X^T e\| \\
&\leq \|P_{U+V}(X^T X r^i - r^i)\| + \|P_U r^i\| + \sqrt{1+\delta}\|e\| \qquad (11) \\
&= \|P_{U+V} X^T X P_{U+V} r^i - r^i\| + \|P_U r^i\| + \sqrt{1+\delta}\|e\| \qquad (12) \\
&\leq \delta\|r^i\| + \|P_U r^i\| + \sqrt{1+\delta}\|e\| . \qquad (13)
\end{aligned}
$$

Equation (11) uses Lemma 12 and $U \subseteq U + V$, which implies that projecting onto the subspace $U + V$ instead of $U$ cannot decrease the norm. Equation (12) follows from the definition of $V$, which implies $r^i \in V$ and hence $P_{U+V} r^i = r^i$. Equation (13) uses Lemma 12 again.

Combining Equations (10) and (13) gives

$$
\|P_U r^i\| \geq \eta_0 \|r^i\| - \rho_0 \|e\| ,
$$

where $\eta_0$ and $\rho_0$ are as defined in the statement of the lemma. Since we also have $\|P_{U^\perp} r^i\|^2 = \|r^i\|^2 - \|P_U r^i\|^2$, we can now establish the desired upper bound. To simplify notation, we complete our proof with the following claim.

**Claim 14.** *Let* $w, x, y, z \in \mathbb{R}$ *be such that* $x \geq \eta_0 z - w$ *and* $y^2 = z^2 - x^2$. *Then*

$$
y \leq \sqrt{1 - \eta_0^2}\, z + \frac{\eta_0 w}{\sqrt{1 - \eta_0^2}} .
$$

Instantiating Lemma 14 with $w = \rho_0 \|e\|$, $x = \|P_U r^i\|$, $y = \|r^i\|$, and $z = \|P_{U^\perp} r^i\|$ then directly implies Equation (8). So it only remains to prove Claim 14, which we accomplish by completing the square:

$$
\begin{aligned}
y^2 &= z^2 - x^2 \\
&\leq z^2 - (\eta_0 z - w)^2 \\
&= (1 - \eta_0^2) z^2 + 2\eta_0 z w - w^2 \\
&= (1 - \eta_0^2) z^2 + 2\eta_0 z w + \frac{\eta_0^2 w^2}{1 - \eta_0^2} - \frac{\eta_0^2 w^2}{1 - \eta_0^2} - w^2 \\
&= \left( \sqrt{1 - \eta_0^2}\, z + \frac{\eta_0 w}{\sqrt{1 - \eta_0^2}} \right)^2 - \frac{\eta_0^2 w^2}{1 - \eta_0^2} - w^2 .
\end{aligned}
$$

Since $\eta_0 < 1$, this proves Claim 14. $\qquad\square$

Next, we prove that the iterates $\hat{\theta}^i$ of AS-IHT converge linearly.

**Theorem 9.** *Let* $\hat{\theta}^i$ *be the estimate computed by* AS-IHT *in iteration* $i$ *and let* $r^{i+1} = \theta^* - \hat{\theta}^{i+1}$ *be the corresponding residual. Moreover, let* $\mathbb{U}$ *be an arbitrary subspace model. We also assume:*
- $y = X\theta^* + e$ *as in Equation* (1) *with* $\theta^* \in \mathcal{M}(\mathbb{U})$.
- $\mathcal{T}$ *is a* $(c_\mathcal{T}, \mathbb{U}, \mathbb{U}_\mathcal{T})$-*approximate tail projection.*
- $\mathcal{H}$ *is a* $(c_\mathcal{H}, \mathbb{U} \oplus \mathbb{U}_\mathcal{T}, \mathbb{U}_\mathcal{H})$-*approximate head projection.*
- *The matrix* $X$ *satisfies the* $(\mathbb{U} \oplus \mathbb{U}_\mathcal{T} \oplus \mathbb{U}_\mathcal{H}, \delta)$-*subspace RIP.*

*Then the residual error of the next iterate, i.e.,* $r^{i+1} = \theta^* - \hat{\theta}^{i+1}$ *satisfies*

$$
\|r^{i+1}\| \leq \eta\|r^i\| + \rho\|e\| ,
$$

*where* $\quad \eta = (1 + c_\mathcal{T})\left( \delta + \sqrt{1 - \eta_0^2} \right) , \qquad\qquad \rho = (1 + c_\mathcal{T})\left( \frac{\eta_0 \rho_0}{\sqrt{1 - \eta_0^2}} + \sqrt{1+\delta} \right) ,$

$\qquad\qquad \eta_0 = c_\mathcal{H}(1 - \delta) - \delta , \qquad\qquad\qquad and \qquad \rho_0 = (1 + c_\mathcal{H})\sqrt{1+\delta} .$

*Proof.* We start by applying the tail projection property of $\mathcal{T}$ on the input vector $a = \hat{\theta}^i + \mathcal{H}(b^i)$:

$$
\begin{aligned}
\left\| r^{i+1} \right\| = \left\| \theta^* - \hat{\theta}^{i+1} \right\| &= \left\| \theta^* - T(a) \right\| \\
&\leq \| \theta^* - a \| + \| a - T(a) \| \\
&\leq (1 + c_\mathcal{T}) \| \theta^* - a \| \\
&= (1 + c_\mathcal{T}) \left\| r^i - \mathcal{H}(X^T X r^i + X^T e) \right\| .
\end{aligned}
\tag{14}
$$

Intuitively, the quantity on the right hand side of (14) is small for two reasons: first, the matrix $X^T X$ behaves close to an isometry on the vector $r^i$ because $r^i$ is in the subspace model $\mathbb{U} \oplus \mathbb{U}_\mathcal{T}$. Second, as we have shown in Lemma 13, the subspace identified by the approximate head projection $\mathcal{H}$ captures a good fraction of the residual $r^i$, and hence $r^i - \mathcal{H}(b^i)$ is small.

More formally, let the subspaces $U$ and $V$ be defined as before in Lemma 13, i.e., $U = \mathcal{H}(b^i)$ and $V \in \mathbb{U} \oplus \mathbb{U}_\mathcal{T}$ is such that $r^i \in V$. Then we get

$$
\begin{aligned}
\left\| r^i - H(X^T X r^i + X^T e) \right\| &= \left\| P_U r^i + P_{U^\perp} r^i - P_U X^T X r^i - P_U X^T e \right\| \\
&\leq \left\| P_U (X^T X r^i - r^i) \right\| + \left\| P_{U^\perp} r^i \right\| + \left\| P_U X^T e \right\| \\
&\leq \left\| P_{U+V} (X^T X r^i - r^i) \right\| + \left\| P_{U^\perp} r^i \right\| + \left\| P_U X^T e \right\| \\
&= \left\| P_{U+V} X^T X P_{U+V} r^i - r^i \right\| + \left\| P_{U^\perp} r^i \right\| + \left\| P_U X^T e \right\| \quad (15) \\
&\leq \delta \| r^i \| + \| P_{U^\perp} r^i \| + \sqrt{1 + \delta} \| e \| . \tag{16}
\end{aligned}
$$

Equation (15) uses that $r^i \in V$, and in Equation (16) we invoke consequences of the RIP (see Lemma 12). Combining Equations (14), (16), and Lemma 13 then gives

$$
\left\| r^{i+1} \right\| \leq (1 + c_\mathcal{T}) \left( \delta \| r^i \| + \sqrt{1 - \eta_0^2} \| r^i \| + \frac{\eta_0 \rho_0}{\sqrt{1 - \eta_0^2}} \| e \| + \sqrt{1 + \delta} \| e \| \right) .
$$

Rearranging this inequality yields the statement of the theorem. $\qquad \square$

While Theorem 9 only gives a guarantee from one iteration of AS-IHT to the next, it is straightforward to extend this to a guarantee for the entire algorithm.

**Corollary 15.** *We adopt the setting of Theorem 9, i.e.,*
- *$y = X\theta^* + e$ as in Equation (1) with $\theta^* \in \mathcal{M}(\mathbb{U})$.*
- *$\mathcal{T}$ is a $(c_\mathcal{T}, \mathbb{U}, \mathbb{U}_\mathcal{T})$-approximate tail projection.*
- *$\mathcal{H}$ is a $(c_\mathcal{H}, \mathbb{U} \oplus \mathbb{U}_\mathcal{T}, \mathbb{U}_\mathcal{H})$-approximate head projection.*
- *The matrix $X$ satisfies the $(\mathbb{U} \oplus \mathbb{U}_\mathcal{T} \oplus \mathbb{U}_\mathcal{H}, \delta)$-subspace RIP.*

*Furthermore, assume that $c_\mathcal{T}$, $c_\mathcal{H}$, and $\delta$ are such that $\eta < 1$. Set the number of iterations to*

$$
t = \left\lceil \frac{\log \frac{\| \theta^* \|}{\| e \|}}{\log \frac{1}{\eta}} \right\rceil .
$$

*Then* AS-IHT$(y, X, t)$ *returns an estimate $\hat{\theta}$ such that*

$$
\| \theta^* - \hat{\theta} \| \leq \left( 1 + \frac{\rho}{1 - \eta} \right) \| e \| . \tag{17}
$$

*Proof.* Note that $\left\| r^0 \right\| = \| \theta^* \|$ due to our initialization $\hat{\theta}^0 = 0$. Invoking Theorem 9 and a straightforward induction then yields

$$
\| \theta^* - \hat{\theta} \| = \left\| \theta^* - \hat{\theta}^{t+1} \right\| = \left\| r^{t+1} \right\| = \eta^t \| \theta^* \| + \rho \| e \| \sum_{i=0}^t \eta^i .
$$

We can bound the first term on the RHS because we have $\eta^t \| \theta^* \| \leq \| e \|$ for $t$ as defined above. In the second term on the RHS, we bound the geometric series by $\frac{1}{1-\eta}$. Combining these bounds yields Equation (17). $\qquad \square$

Note that Corollary 15 is essentially the formal version of Theorem 5 stated in Section 2. For completeness, we first repeat Theorem 5:

**Theorem 5** (informal). *Let $\mathcal{H}$ and $\mathcal{T}$ be approximate head and tail projections with constant approximation ratios, and let the matrix $X$ satisfy the $(\oplus^c \mathbb{U}, \delta)$-subspace RIP for a sufficiently large constant $c$ and a sufficiently small constant $\delta$. Then there is an algorithm AS-IHT that returns an estimate $\hat{\theta}$ such that $\|\hat{\theta} - \theta^*\| \leq C\|e\|$. The algorithm requires $O(\log\|\theta\|/\|e\|)$ multiplications with $X$ and $X^T$, and $O(\log\|\theta\|/\|e\|)$ invocations of $\mathcal{H}$ and $\mathcal{T}$.*

Let $c_1$ and $c_2$ be fixed constants. When $\mathcal{T}$ is a $(c_{\mathcal{T}}, \mathbb{U}, \oplus^{c_1} \mathbb{U})$-approximate tail projection and $\mathcal{H}$ is a $(c_{\mathcal{H}}, \oplus^{c_1} \mathbb{U} \oplus \mathbb{U}, \oplus^{c_2} \mathbb{U})$-approximate head projection, Theorem 5 is the special case where

- $\mathbb{U}_{\mathcal{T}} = \oplus^{c_1} \mathbb{U}$
- $\mathbb{U}_{\mathcal{H}} = \oplus^{c_2} \mathbb{U}$
- $c = 1 + c_1 + c_2$.

The iteration bound from Corollary 15 implies the bound on the number of multiplications with $X$ and $X^T$, and the bound on the number of invocations of $\mathcal{H}$ and $\mathcal{T}$.

## A.2 Boosting approximate projections

In some cases, it is hard to design efficient approximate projection algorithms that satisfy the stringent conditions on $c_{\mathcal{T}}$ and $c_{\mathcal{H}}$ in Theorem 9. To overcome this difficulty, we now show how to "boost" the approximation ratio of an approximate head projection to be arbitrarily close to 1.

First, we start with a single iteration of boosting.

**Theorem 16.** *Let $\mathcal{H} : \mathbb{R}^d \to \mathbb{U}_{\mathcal{H}}$ be a $(c_{\mathcal{H}}, \mathbb{U}, \mathbb{U}_{\mathcal{H}})$-approximate head projection running in time $O(T)$. Then we can construct a $(\sqrt{2c_{\mathcal{H}}^2 - 2c_{\mathcal{H}}^3 + c_{\mathcal{H}}^4}, \mathbb{U}, \mathbb{U}_{\mathcal{H}} \oplus \mathbb{U}_{\mathcal{H}})$-approximate head projection running in time $O(T + T_1' + T_2')$, where $T_1'$ is the time needed to apply a projection onto a subspace in $\mathbb{U}_{\mathcal{H}}$, and $T_2'$ is the time needed to find an orthogonal projector for the sum of two subspaces in $\mathbb{U}_{\mathcal{H}}$.*

*Proof.* Consider Algorithm 2. The running time bound follows directly from the definition of BOOSTEDHEAD1. It is also easy to see that the returned subspace is in $\mathbb{U}_{\mathcal{H}} \oplus \mathbb{U}_{\mathcal{H}}$. Hence it remains to show that BOOSTEDHEAD1 satisfies the desired approximation ratio.

In the following, let $OPT = \max_{U' \in \mathbb{U}} \|P_{U'}b\|^2$ be the best possible head approximation, and let $U^* \in \mathbb{U}$ be a subspace achieving $OPT$. Moreover, let $\tilde{c}_{\mathcal{H}}$ be the head-approximation ratio achieved by the subspace $U$, i.e.,

$$\|P_U b\|^2 = \tilde{c}_{\mathcal{H}}^2 OPT .$$

Let $W$ be the subspace returned by the algorithm. Then we have

$$\|P_W b\|^2 = \|P_U P_W b\|^2 + \|P_{U^\perp} P_W b\|^2 . \tag{18}$$

We can write $P_U = B^T B$ for an orthogonal basis $B$ of the subspace $U$, and $P_W = [B^T | D^T] \left[ \begin{smallmatrix} B \\ D \end{smallmatrix} \right]$, where $D$ is an orthonormal basis of the orthogonal complement of $U$ in $W$ (it is easy to see that such a pair of bases always exists, e.g., by following the Gram-Schmidt procedure). Basic linear algebra then shows that $P_U P_W = P_U = P_W P_U$. We can use this fact to bound the first term above:

$$\|P_U P_W b\|^2 = \|P_U b\|^2 = \tilde{c}_{\mathcal{H}}^2 OPT . \tag{19}$$

Next, we consider the second term in Equation (18). We have

$$P_{U^\perp} P_W = (I - P_U)P_W = P_W - P_U P_W = P_W - P_W P_U = P_W(I - P_U) .$$

Since $(I - P_U)b = r$, this gives

$$\|P_{U^\perp} P_W b\|^2 = \|P_W(I - P_U)b\|^2 = \|P_W r\|^2 \geq \|P_V r\|^2 , \tag{20}$$

where the last equality follows from the fact that the subspace $W$ contains the subspace $V$.

From the head-approximation guarantee of the oracle $\mathcal{H}$, we know that

$$\|P_V r\|^2 \geq c_{\mathcal{H}}^2 \max_{U' \in \mathbb{U}} \|P_{U'} r\|^2 \geq c_{\mathcal{H}}^2 \|P_{U^*} r\|^2 . \tag{21}$$

Next, we bound $\|P_{U^*}r\|$ (note that we omitted the square).

$$
\begin{aligned}
\|P_{U^*}r\| \ = \ \|P_{U^*}(I - P_U)b\| \ = \ & \|P_{U^*}b - P_{U^*}P_Ub\| \\
\geq \ & \|P_{U^*}b\| - \|P_{U^*}P_Ub\| \\
\geq \ & \|P_{U^*}b\| - \|P_Ub\| \\
= \ & \sqrt{OPT} - \tilde{c}_{\mathcal{H}}\sqrt{OPT} \\
= \ & (1 - \tilde{c}_{\mathcal{H}})\sqrt{OPT} \ .
\end{aligned}
$$

The second line uses the triangle inequality, the third line uses the fact that $P_U$ is an orthogonal projection, and the fourth line uses the optimality of the subspace $U^*$ and the approximation guarantee of the subspace $U$, respectively. Squaring both sides then yields

$$
\|P_{U^*}r\|^2 \ \geq \ (1 - \tilde{c}_{\mathcal{H}})^2 \, OPT \ . \tag{22}
$$

We can now combine Equations (18) to (22) and get

$$
\begin{aligned}
\|P_W b\|^2 \ \geq \ & \tilde{c}_{\mathcal{H}}^2 \, OPT + c_{\mathcal{H}}^2 \, (1 - \tilde{c}_{\mathcal{H}})^2 \, OPT \\
= \ & (\tilde{c}_{\mathcal{H}}^2 + c_{\mathcal{H}}^2 - 2c_{\mathcal{H}}^2\tilde{c}_{\mathcal{H}} + c_{\mathcal{H}}^2\tilde{c}_{\mathcal{H}}^2) OPT \ . \tag{23}
\end{aligned}
$$

We know that $c_{\mathcal{H}} \leq \tilde{c}_{\mathcal{H}} \leq 1$. In order to get a uniform bound, we analyze the factor in front of $OPT$. Let $x = \tilde{c}_{\mathcal{H}}^2$, then we can write the approximation ratio as

$$
f(x) \ = \ (1 + c_{\mathcal{H}}^2)x^2 - 2c_{\mathcal{H}}^2 x + c_{\mathcal{H}}^2 \ .
$$

Computing the derivative and setting it to zero yields

$$
\begin{aligned}
f'(x) \ = \ & 2(1 + c_{\mathcal{H}}^2)x - 2c_{\mathcal{H}}^2 \\
x \ = \ & \frac{c_{\mathcal{H}}^2}{1 + c_{\mathcal{H}}^2} \ .
\end{aligned}
$$

So the unconstrained minimum is achieved for some value of $x \leq c_{\mathcal{H}}^2$. Since the quadratic function $f$ is increasing for $x \geq c_{\mathcal{H}}^2$, the constrained minimum is achieved for $\tilde{c}_{\mathcal{H}} = c_{\mathcal{H}}$, which gives

$$
\|P_W b\|^2 \ \geq \ (2c_{\mathcal{H}}^2 - 2c_{\mathcal{H}}^3 + c_{\mathcal{H}}^4)OPT \ . \qquad \square
$$

---

**Algorithm 2** Boosted head projection

1: **function** BOOSTEDHEAD1($\mathcal{H}, b$)
2:     $U \leftarrow \mathcal{H}(b)$
3:     $r \leftarrow b - P_U b$
4:     $V \leftarrow \mathcal{H}(r)$
5:     **return** an orthogonal projection onto the subspace $U + V$

6: **function** BOOSTEDHEAD($\mathcal{H}, b, t$)
7:     **if** $t = 1$ **then**
8:         **return** $\mathcal{H}(b)$
9:     **else**
10:         **return** BOOSTEDHEAD1(BOOSTEDHEAD($\mathcal{H}, \cdot, t - 1$), $b$)

---

Next, we extend one iteration of boosting to several rounds. In our final applications of head approximation boosting, we are mainly interested in boosting a constant head approximation ratio $c_{\mathcal{H}}$ to an improved head approximation ratio $c'_{\mathcal{H}}$ that is close to one but still a constant. Hence it suffices to state a boosting result without explicit dependence between $c_{\mathcal{H}}$ and $c'_{\mathcal{H}}$, which simplifies the argument in the following theorem.

**Theorem 10.** *Let $\mathcal{H}$ be a $(c_{\mathcal{H}}, \mathbb{U}, \mathbb{U}_{\mathcal{H}})$-approximate head projection running in time $O(T)$, and let $\varepsilon > 0$. Then there is a constant $c = c_{\varepsilon, c_{\mathcal{H}}}$ that depends only on $\varepsilon$ and $c_{\mathcal{H}}$ such that we can construct a $(1 - \varepsilon, \mathbb{U}, \oplus^c \mathbb{U}_{\mathcal{H}})$-approximate head projection running in time $O(c(T + T'_1 + T'_2))$ where $T'_1$ is the time needed to apply a projection onto a subspace in $\oplus^c \mathbb{U}_{\mathcal{H}}$, and $T'_2$ is the time needed to find an orthogonal projector for the sum of two subspaces in $\oplus^c \mathbb{U}_{\mathcal{H}}$.*

*Proof.* Consider the algorithm BOOSTEDHEAD. If BOOSTEDHEAD$(\mathcal{H}, b, t)$ achieves head approximation ratio $c_{\mathcal{H},t}$, then BOOSTEDHEAD$(\mathcal{H}, b, t+1)$ achieves head approximation ratio $c_{\mathcal{H},t+1} = \sqrt{2c_{\mathcal{H},t}^2 - 2c_{\mathcal{H},t}^3 + c_{\mathcal{H},t}^4}$ (see Theorem 16). Hence it suffices to show that the sequence $c_{\mathcal{H},t}$ converges to 1 for any starting value $c_{\mathcal{H},0} = c_{\mathcal{H}}$.

Consider the function $f(x) = \sqrt{2x^2 - 2x^3 + x^4}$ and note that $c_{\mathcal{H},t+1} = f(c_{\mathcal{H},t})$. An elementary calculation shows that $f(x) > x$ for $0 < x < 1$. Hence the sequence $x_{i+1} = f(x_i)$ converges to 1 for any $0 < x_0 < 1$. For a proof by contradiction, let $x' < 1$ be the smallest value such that $x_i \le x'$ for all $i$. Let $\delta = f(x') - x' > 0$. Since $f$ is continuous, we can find a point $x_{i^*}$ close to $x'$ such that $f(x_{i^*}) > f(x') - \delta$ and hence $f(x_{i^*}) > x'$, a contradiction. So for any $\varepsilon > 0$, there is a $c = c_{\varepsilon,c_{\mathcal{H}}}$ such that $c_{\mathcal{H},c} \ge 1 - \varepsilon$. □

### A.3 A boosted recovery algorithm

We now combine our convergence result for AS-IHT with the boosting technique to prove a general result that holds for *any* constant head an tail approximation ratios.

**Theorem 17.** *We make the following assumptions:*
- $y = X\theta^* + e$ *as in Equation* (1) *with* $\theta^* \in \mathcal{M}(\mathbb{U})$.
- $\mathcal{T}$ *is a* $(c_{\mathcal{T}}, \mathbb{U}, \mathbb{U}_{\mathcal{T}})$-*approximate tail projection.*
- $\mathcal{H}$ *is a* $(c_{\mathcal{H}}, \mathbb{U} \oplus \mathbb{U}_{\mathcal{T}}, \mathbb{U}_{\mathcal{H}})$-*approximate head projection.*
- *The matrix* $X$ *satisfies the* $(\oplus^c (\mathbb{U} \oplus \mathbb{U}_{\mathcal{T}} \oplus \mathbb{U}_{\mathcal{H}}), \delta)$-*subspace RIP for* $c$ *sufficiently large and* $\delta$ *sufficiently small, where* $c$ *and* $\delta$ *depend only on* $c_{\mathcal{T}}$ *and* $c_{\mathcal{H}}$.

*Then there is an algorithm* BOOSTED-AS-IHT *that returns an estimate* $\hat{\theta}$ *such that*

$$\|\theta^* - \hat{\theta}\| \le C\|e\|,$$

*where* $C$ *depends only on* $\delta$, $c_{\mathcal{T}}$, *and* $c_{\mathcal{H}}$. *Moreover, the algorithm requires* $O(\log \|\theta\|/\|e\|)$ *iterations.*

*Proof.* The algorithm BOOSTED-AS-IHT is AS-IHT (Algorithm 1) with BOOSTEDHEAD (Algorithm 2) in place of the approximate head projection.

In order to invoke Corollary 15, we need to show that

$$\eta = (1 + c_{\mathcal{T}}) \left( \delta + \sqrt{1 - \eta_0^2} \right)$$

is less than 1, where $\eta_0$ is given by (c.f. Theorem 9)

$$\eta_0 = c_{\mathcal{H}}(1 - \delta) - \delta.$$

Note that by making $\delta$ sufficiently small and $c_{\mathcal{H}}$ sufficiently close to 1, we can achieve $\eta_0$ arbitrarily close to 1 and hence $\eta$ arbitrarily small for any *fixed* tail approximation ratio $c_{\mathcal{T}}$.

While the assumption in the current theorem allows for small $\delta$ as long as $\delta$ only depends on $c_{\mathcal{H}}$ and $c_{\mathcal{T}}$, we need to handle arbitrarily small, fixed $c_{\mathcal{H}}$. In order to do so, we invoke Theorem 10, which allows us to get a boosted head approximation with approximation ratio $c'_{\mathcal{H}}$ arbitrarily close to 1. The invocation of BOOSTEDHEAD leads to a larger output set $\oplus^c \mathbb{U}_{\mathcal{H}}$. As a result, we require the subspace-RIP over the set $\mathbb{U} \oplus \mathbb{U}_{\mathcal{T}} \oplus \oplus^c \mathbb{U}_{\mathcal{H}}$. The current theorem provides this subspace-RIP by assumption. □

## B Proofs for low-rank matrix recovery

We now formally show how to convert the approximate SVD guarantees of [17] to approximate head and tail projections for low-rank matrices. For convenience, we first repeat the main result of [17].

**Fact 11** ([17]). *There is an algorithm* APPROXSVD *with the following guarantee. Let* $A \in \mathbb{R}^{d_1 \times d_2}$ *be an arbitrary matrix, let* $r \in \mathbb{N}$ *be the target rank, and let* $\varepsilon > 0$ *be the desired accuracy. Then with probability* $1 - \psi$, APPROXSVD$(A, r, \varepsilon)$ *returns an orthonormal set of vectors* $z_1, \ldots, z_r \in \mathbb{R}^{d_1}$ *such that for all* $i \in [r]$, *we have*

$$\left| z_i^T A A^T z_i - \sigma_i^2 \right| \le \varepsilon \sigma_{r+1}^2, \tag{3}$$

*where $\sigma_i$ is the $i$-th largest singular value of A. Furthermore, let $Z \in \mathbb{R}^{d_1 \times r}$ be the matrix with columns $z_i$. Then we also have*

$$\left\| A - ZZ^T A \right\|_F \ \leq \ (1+\varepsilon)\|A - A_r\|_F \ , \tag{4}$$

*where $A_r$ is the best rank-r Frobenius-norm approximation of A. Finally, the algorithm runs in time $O\left( \frac{d_1 d_2 r \log(d_2/\psi)}{\sqrt{\varepsilon}} + \frac{d_1 r^2 \log^2(d_2/\psi)}{\varepsilon} + \frac{r^3 \log^3(d_2/\psi)}{\varepsilon^{3/2}} \right).$*

As mentioned before, Equation (4) directly gives a tail approximation. We now show how to convert Equation (3) to a head approximation guarantee. In the following, we let $\mathbb{U}_r$ be the subspace model of rank-$r$ matrices.

**Theorem 18.** *There is an algorithm* APPROXLOWRANK *with the following property. For an arbitrary input matrix $A \in \mathbb{R}^{d_1 \times d_2}$ and a target rank r,* APPROXLOWRANK *produces a subspace of rank-r matrices U and a matrix $Y = P_U A$, the projection of A onto U. With probability 99/100, the output satisfies both an $(1 - \varepsilon, \mathbb{U}_r, \mathbb{U}_r)$-approximate head projection guarantee and an $(1 + \varepsilon, \mathbb{U}_r, \mathbb{U}_r)$- approximate tail projection guarantee. Moreover,* APPROXLOWRANK *runs in time*

$$O\left( \frac{d_1 d_2 r \log d_2}{\sqrt{\varepsilon}} + \frac{d_1 r^2 \log^2 d_2}{\varepsilon} + \frac{r^3 \log^3 d_2}{\varepsilon^{3/2}} \right) \ .$$

*Proof.* Let $z_1, \ldots, z_r$ be the vectors returned by APPROXLOWRANK$(A, r, \varepsilon)$. Then APPROX-LOWRANK returns the matrix $Y = ZZ^T A$ and and the subspace U spanned by the vectors $z_i$ and $z_i^T A$ (it is easy to see that Y is indeed the projection of A onto U). Both operations can be performed in time $O(d_1 d_2 r)$. Hence the overall running time is dominated by the invocation of APPROXSVD, which leads to the running time stated in the theorem.

It remains to prove the desired head and tail approximation ratios. The tail approximation guarantee follows directly from Equation (4). For the head approximation, first note that Equation (3) implies

$$z_i^T A A^T z_i \ \geq \ (1 - \varepsilon)\sigma_i^2 \ .$$

We now apply this inequality by rewriting the head quantity $\left\| ZZ^T A \right\|_F^2$ as follows:

$$
\begin{aligned}
\left\| ZZ^T A \right\|_F^2 \ &= \ \mathrm{tr}(A^T ZZ^T ZZ^T A) \\
&= \ \mathrm{tr}(A^T ZZ^T A) \\
&= \ \mathrm{tr}\left( A^T \left( \sum_{i=1}^{r} z_i z_i^T \right) A \right) \\
&= \ \mathrm{tr}\left( \sum_{i=1}^{r} A^T z_i z_i^T A \right) \\
&= \ \sum_{i=1}^{r} \mathrm{tr}(A^T z_i z_i^T A) \\
&= \ \sum_{i=1}^{r} \mathrm{tr}(z_i^T A A^T z_i) \\
&= \ \sum_{i=1}^{r} z_i^T A A^T z_i \\
&\geq \ (1 - \varepsilon) \sum_{i=1}^{r} \sigma_i^2 \\
&= \ (1 - \varepsilon)\|A_r\|_F^2
\end{aligned}
$$

where the matrix $A_r$ is the best rank-$r$ approximation of the matrix A. This proves the desired head approximation guarantee. $\qquad\square$

### B.1 The final recovery algorithm

We now prove our overall result for low-rank matrix recovery.

**Theorem 6.** *Let $X \in \mathbb{R}^{n \times d}$ be a matrix with subspace-RIP for low-rank matrices, and let $T_X$ denote the time to multiply a $d$-dimensional vector with $X$ or $X^T$. Then there is an algorithm that recovers an estimate $\hat{\theta}$ such that $\|\hat{\theta} - \theta^*\| \leq C\|e\|$. Moreover, the algorithm runs in time $\widetilde{O}(T_X + r \cdot d_1^2)$.*

*Proof.* We assume that $X$ satisfies the low-rank RIP for matrices of rank $4r$ and RIP constant $\delta \leq 0.1$. We remark that it is possible to fine-tune these constants, but our focus here is on the scaling with the problem dimensions.

Instantiating Theorem 18 gives us approximate head and tail projections with the following guarantees:

- $\mathcal{T}$ is a $(1.1, \mathbb{U}_r, \mathbb{U}_r)$-approximate tail projection.
- $\mathcal{H}$ is a $(0.9, \mathbb{U}_{2r}, \mathbb{U}_{2r})$-approximate tail projection.

Note that $\mathbb{U}_r \oplus \mathbb{U}_r \subseteq \mathbb{U}_{2r}$, so $\mathcal{T}$ and $\mathcal{H}$ satisfy the conditions of Theorem 9. Moreover, $\mathbb{U} \oplus \mathbb{U}_{\mathcal{T}} \oplus \mathbb{U}_{\mathcal{H}} \subseteq \mathbb{U}_{4r}$, and therefore the matrix $X$ also satisfies the RIP condition of Theorem 9. Substituting $c_{\mathcal{T}} = 1.1$, $c_{\mathcal{H}} = 0.9$, and $\delta = 0.1$ into Theorem 9 then yields $\eta < 0.9$, so we can invoke Corollary 15.

Corollary 15 direct implies the desired recovery guarantee $\|\theta^* - \hat{\theta}\| \leq C\|e\|$. Moreover, the corresponding bound on the number of iterations is $O(\log\|\theta^*\|/\|e\|)$. This has two consequences: (i) The total number of multiplications with $X$ or $X^T$ is $\widetilde{O}(1)$. (ii) The total number of invocations of the approximate head and tail projections is $\widetilde{O}(1)$. Recall that each matrix multiplication with $X$ takes $T_X$ time, and that the time complexity of the approximate projections is $\widetilde{O}(r \cdot d_1^2)$, where we again assume the square case for simplicity. Combining these results gives the stated time complexity. $\square$

We remark that for fast design matrices (e.g., structured observations such as a subsampled Fourier matrix), we have $T_X = \widetilde{O}(d_1^2)$ and the total running time becomes $\widetilde{O}(r \cdot d_1^2)$. See Appendix D for such a construction.

## C Approximation algorithms for 2D histograms

We now describe our approximate head and tail projections for histograms. One key ingredient in our algorithms are *hierarchical* histograms. Overall, our goal is to approximate arbitrary 2D histograms, i.e., arbitrary partitions of a $\sqrt{d} \times \sqrt{d}$ matrix with $k$ non-overlapping rectangles (for simplicity, we limit our attention to the case of square matrices). Such histograms are also known as *tiling* histograms. However, tiling histograms are hard to work with algorithmically because they do not allow a clean decomposition for a dynamic program. Instead, work in histogram approximation has utilized hierarchical histograms, which are also partitions of a matrix into $k$ non-overlapping rectangles. The additional restriction is that the partition can be represented as a tree in which each rectangle arises through a vertical or horizontal split of the parent rectangle. We refer the reader to [18] for a more detailed description of different histogram types.

An important result is that every tiling histogram consisting of $k$ rectangles can be simulated with a hierarchical histogram consisting of at most $4k$ rectangles (d'Amore and Franciosa, 1992). Since Theorems 7 and 8 provide bicriterion guarantees for the output space, i.e., projections into a space of histograms consisting of $O(k)$ rectangles, we focus our attention on approximation algorithms for hierarchical histograms in the following. These results can then easily be converted into statements for tiling histograms by increasing the number of histogram tiles by 4.

Next, we introduce some histogram-specific notation. For a histogram subspace $U$, we denote the number of histogram pieces in $U$ with $\gamma(U)$. We denote the set of hierarchical histograms subspaces with $\mathscr{H}_h$. When we have an upper bound on the number of histogram pieces, we write $\mathscr{H}_{h,k}$ for the set of hierarchical histogram subspaces $U$ with $\gamma(U) \leq k$.

An important subroutine in our approximate projections is the following notion of a hierachical histogram oracle.

**Definition 19.** *An $(\alpha, \zeta)$-hierarchical histogram oracle is an algorithm with the following guarantee: given any $b \in \mathbb{R}^{\sqrt{d} \times \sqrt{d}}$ and $\lambda \in \mathbb{R}$ as input, the algorithm returns a hierarchical histogram subspace $U$ such that*

$$\|P_U b\|^2 - \frac{\lambda}{\alpha}\gamma(U) \geq \max_{U' \in \mathscr{H}_h} \|P_{U'}b\|^2 - \lambda\gamma(U') . \tag{24}$$

*Moreover, the algorithm runs in time $O(d^{1+\zeta})$.*

An algorithm with the following guarantee directly follows from the hierarchical dynamic programming techniques introduced in [18]. In particular, Theorem 3 of [18] implies a dependence of $\alpha = O(1/\zeta^2)$.

Equation (24) has the flavor of a head approximation (a $\max$-quantified guarantee). As a direct consequence of Equation (24), we also get the following "tail approximation" variant.

**Lemma 20.** *The solution $U$ returned by an $(\alpha, \zeta)$-hierarchical histogram oracle also satisfies*

$$\|b - P_U b\|^2 + \frac{\lambda}{\alpha}\gamma(U) \leq \min_{U' \in \mathscr{H}_h} \|b - P_{U'}b\|^2 + \lambda\gamma(U') . \tag{25}$$

*Proof.* Multiplying both sides of Equation (24) with $-1$ and pulling the negative sign into the $\max$ gives

$$-\|P_U b\|^2 + \frac{\lambda}{\alpha}\gamma(H) \leq \min_{U' \in \mathscr{H}_h} -\|P_{U'}b\|^2 + \lambda\gamma(U') .$$

Adding $\|b\|^2$ to both sides and using that $P_U$ and $P_{U'}$ are orthogonal projections then gives Equation (25) via the Pythagorean Theorem. $\square$

However, note that neither Equation (24) nor (25) give direct control over the number of histogram pieces $k$. In the following, we give algorithms that convert these guarantees into approximate projections. In a nutshell, we show that carefully choosing the trade-off parameter $\lambda$, combined with a postprocessing step of the corresponding solution, yields head an tail approximations.

## C.1 Approximate tail projection

We now show how to construct an approximate tail projection from a hierarchical histogram oracle. In the following, we assume that $\text{HISTOGRAMORACLE}(b, \lambda)$ is an $(\alpha, \zeta)$-hierarchical histogram oracle.

First, we establish a lower bound on the approximation error $\|b - P_U\|^2$ if $b$ is not in the histogram subspace $U$.

**Lemma 21.** *Let $b \in \mathbb{R}^d$ and $U$ be a histogram subspace. If $b \notin U$, then we have $\|b - P_U\|^2 \geq \varepsilon_{\min}$ where $\varepsilon_{\min}$ is as defined in Algorithm 3.*

*Proof.* If $b \notin U$, there is a histogram piece in $U$ on which $b$ is not constant. Let $R$ be the set of indices in this piece. We now give a lower bound on the projection error based on the histogram piece $R$ (recall that the projection of $b$ onto $U$ averages $b$ in each histogram piece):

$$\|b - P_U\|^2 \geq \sum_{(i,j) \in R} (b_{i,j} - \bar{b}_R)^2 \quad \text{where} \quad \bar{b}_R = \frac{1}{|R|} \sum_{(i,j) \in R} b_{i,j} .$$

Let $(i^*, j^*)$ be the index of the largest coefficient in the histogram piece $R$ (ties broken arbitrarily). Then we bound the sum on the right hand side above with the term corresponding to $(i^*, j^*)$:

$$\|b - P_U\|^2 \geq \left(b_{i^*,j^*} - \frac{1}{|R|} \sum_{(i,j) \in R} b_{i,j}\right)^2 .$$

Let $\Delta_R$ be the smallest non-zero difference between coefficients in $R$. Note that $\Delta_R > 0$ because $b$ is not constant on $R$. Moreover, we have $\Delta_R \leq \max_{(i,j) \in R} b_{i^*,j^*} - b_{i,j}$. Hence we get

$$\left(b_{i^*,j^*} - \frac{1}{|R|} \sum_{(i,j) \in R} b_{i,j}\right)^2 \geq \left(b_{i^*,j^*} - \frac{|R|-1}{|R|}b_{i^*,j^*} - \frac{1}{|R|}(b_{i^*,j^*} - \Delta_R)\right)^2$$

---

**Algorithm 3** Tail projection for hierarchical histograms

---

1: **function** HISTOGRAMTAIL($b, k, \nu, \xi$)
2:      $\Delta \leftarrow \min \left\{ |b_{i,j} - b_{i',j'}| \mid b_{i,j} - b_{i',j'} \neq 0 \right\}$
3:      $\varepsilon_{\min} \leftarrow \frac{\Delta^2}{d^2}$
4:      $\lambda_0 \leftarrow \frac{\varepsilon_{\min}}{2k}$
5:      $U_0 \leftarrow$ HISTOGRAMORACLE($b, \lambda_0$)
6:      **if** $\|b - P_{U_0}\| = 0$ and $\gamma(U_0) \leq \alpha k$ **then**
7:          **return** $U_0$
8:      $\lambda_l \leftarrow 0$
9:      $\lambda_r \leftarrow 2\alpha \|b\|$
10:      $\varepsilon \leftarrow \frac{\varepsilon_{\min}\xi}{k}$
11:      **while** $\lambda_r - \lambda_l \geq \varepsilon$ **do**
12:          $\lambda_m \leftarrow \frac{\lambda_l + \lambda_r}{2}$
13:          $U_m \leftarrow$ HISTOGRAMORACLE($b, \lambda_m$)
14:          **if** $\gamma(U_m) \geq \alpha k$ and $\gamma(U_m) \leq \nu\alpha k$ **then**
15:              **return** $U_m$
16:          **if** $\gamma(U_m) \geq \nu\alpha k$ **then**
17:              $\lambda_l \leftarrow \lambda_m$
18:          **else**
19:              $\lambda_r \leftarrow \lambda_m$
20:      **return** HISTOGRAMORACLE($b, \lambda_r$)

---

because $b_{i^*,j^*}$ is one of the largest coefficients in $R$ and at least one coefficient is smaller than $b_{i^*,j^*}$ by at least $\Delta_R$. Combining the inequalities above and simplifying then yields

$$\|b - P_U\|^2 \geq \frac{\Delta_R^2}{|R|^2} \geq \frac{\Delta^2}{d^2} = \varepsilon_{\min} . \qquad \square$$

Next, we prove that the histogram oracle returns roughly a $k$-histogram if the input is a $k$-histogram and we set the parameter $\lambda$ correctly.

**Lemma 22.** *Let $\varepsilon_{\min}$ and $\lambda_0$ be defined as in Algorithm 3. If $b$ is a hierarchical $k$-histogram, then* HISTOGRAMORACLE($b, \lambda_0$) *returns a hierarchical histogram subspace $U_0$ such that $b \in U_0$ and $\gamma(U_0) \leq \alpha k$.*

*Proof.* First, we show that $b \in U_0$, i.e., that $\|b - P_{U_0}\| = 0$. Since $b \in \mathscr{H}_{\mathrm{h},k}$, we know that there is a hierarchical histogram subspace $U'$ such that $\|b - P_{U'}\| = 0$ and $\gamma(U') \leq k$. Substituting this histogram subspace $U'$ and $\lambda_0$ into Equation (25) gives

$$\|b - P_{U_0}\|^2 \leq \lambda_0 \gamma(U') \leq \frac{\varepsilon_{\min}^2}{2}$$

where we also used that $\frac{\lambda_0}{\alpha}\gamma(U_0) \geq 0$. Since $\varepsilon_{\min} > 0$, the contrapositive of Lemma 21 shows that $b \in U_0$.

Next, we prove that $\gamma(U_0) \leq \alpha k$. Substituting into Equation (25) again and using $\|b - P_{U_0}\| = 0$ now gives the desired bound on the number of histogram pieces:

$$\frac{\lambda_0}{\alpha}\gamma(U_0) \leq \lambda_0 k . \qquad \square$$

With these preliminaries in place, we now show the main result for our tail approximation algorithm.

**Theorem 23.** *Let $b \in \mathbb{R}^d$, $k \in \mathbb{N}$, $\nu > 1$, and $\xi > 0$. Then* HISTOGRAMTAIL($b, k, \nu, \xi$) *returns a histogram subspace $U$ such that $\gamma(U) \leq \nu\alpha k$ and*

$$\|b - P_U b\|^2 \leq \left(1 + \frac{1}{\nu - 1} + \xi\right) \min_{U' \in \mathscr{H}_{\mathrm{h},k}} \|b - P_{U'} b\|^2 .$$

*Moreover, the algorithm runs in time*

$$O\left(n^{1+\zeta} \log\left(\frac{\alpha d \|b\|}{\xi \Delta}\right)\right)$$

*where $\Delta$ is as defined in Algorithm 3.*

*Proof.* We analyze the three cases in which HISTOGRAMTAIL returns separately. First, consider Line 7. In this case, $U_0$ clearly satisfies the conditions of the theorem. So in the following, we condition on the algorithm not returning in Line 7. By the contrapositive of Lemma 22, this implies that $b \notin \mathcal{M}(\mathscr{H}_{\mathrm{h},k})$.

Next, consider the case that HISTOGRAMTAIL returns in Line 15. This directly implies that $\gamma(U_m) \le \nu\alpha k$. Moreover, substituting into Equation 25 and restricting the right hand side to histogram subspaces with at most $k$ pieces gives

$$\|b - P_{U_m}\|^2 + \frac{\lambda_m}{\alpha}\gamma(U_m) \le \min_{U' \in \mathscr{H}_{\mathrm{h},k}} \|b - P_{U'}b\|^2 + \lambda_m\gamma(U')$$

$$\|b - P_{U_m}\|^2 \le \min_{U' \in \mathscr{H}_{\mathrm{h},k}} \|b - P_{U'}b\|^2 + \lambda_m\gamma(U') - \lambda_m k$$

$$\|b - P_{U_m}\|^2 \le \min_{U' \in \mathscr{H}_{\mathrm{h},k}} \|b - P_{U'}b\|^2$$

where we used that $\gamma(U_m) \ge \alpha k$ and $\gamma(U') \le k$.

For the remaining case (Line 20), we use the following shorthands in order to simplify notation: Let $U_l$ and $U_r$ be the histogram subspaces returned by HISTOGRAMORACLE with parameters $\lambda_l$ and $\lambda_r$, respectively. We denote the corresponding tail errors with $t_l = \|b - P_{U_l}\|^2$ and $t_r = \|b - P_{U_r}\|^2$. Moreover, we denote the optimal tail error with $t^* = \min_{U' \in \mathscr{H}_{\mathrm{h},k}} \|b - P_{U'}b\|^2$. Finally, let $\gamma_l = \gamma(U_l)$ and $\gamma_r = \gamma(U_r)$ be the number of histogram pieces in the respective histogram subspaces. Rewriting Equation 25 in terms of the new notation gives

$$t_l + \frac{\lambda_l}{\alpha}\gamma_l \le t^* + \lambda_l k \tag{26}$$

$$t_r + \frac{\lambda_r}{\alpha}\gamma_r \le t^* + \lambda_r k \tag{27}$$

We will use Equation 26 in order to bound our tail projection error $t_l$. For this, we establish an upper bound on $\lambda_r$. Note that $\lambda_r \le \lambda_l + \varepsilon$ when the algorithm reaches Line 20. Moreover, the binary search over $\lambda$ is initialized so that we always have $\gamma_l > \nu\alpha k$ and $\gamma_r < \alpha k$. Combining these facts with Equation (27) leads to an upper bound on $\lambda_l$:

$$t_l + \frac{\lambda_l}{\alpha}\gamma_l \le t^* + \lambda_l k$$

$$\frac{\lambda_l}{\alpha}\nu\alpha k \le t^* + \lambda_l k$$

$$\lambda_l \le \frac{t^*}{(\nu - 1)k} \ .$$

We use these facts in order to establish an upper bound on $t_r$. Substituting into Equation (27) gives

$$t_r \le t^* + (\lambda_l + \varepsilon)k$$

$$t_r \le t^* + \frac{t^*}{\nu - 1} + \frac{\varepsilon_{\min}\xi}{k}k$$

$$t_r \le \left(1 + \frac{1}{\nu - 1} + \xi\right)t^*$$

where we used that $t^* \ge \varepsilon_{\min}$ because $b$ is not a hierarchical $k$-histogram if the algorithm reaches Line 20 (see Lemma 21). Combined with the fact that $\gamma_r \le \alpha k$, this proves the statement of the theorem.

Finally, we consider the running time bound. It is straightforward to see that the overall running time is dominated by the invocations of HISTOGRAMORACLE, each of which takes $O(d^\varsigma)$ time. The number of iterations of the binary search is bounded by the initial gap between $\lambda_l$ and $\lambda_r$ and the final gap $\varepsilon$, which gives an iteration bound of

$$\left\lceil \log \frac{\lambda_r^{(0)} - \lambda_l^{(0)}}{\varepsilon} \right\rceil = O\left(\log\left(\frac{\alpha d^2 k\|b\|}{\xi\Delta^2}\right)\right) \ .$$

---
**Algorithm 4** Head projection for hierarchical histograms
---
1: **function** HISTOGRAMHEAD($b, k, \tau$)
2:      $b_{\max} \leftarrow \max_{b_{i,j}} \left| b_{i,j}^2 \right|$
3:      $\lambda_l \leftarrow \frac{b_{\max}\tau}{k}$
4:      $U_l \leftarrow$ HISTOGRAMORACLE($b, \lambda_l$)
5:      **if** $\gamma(U_l) \leq \frac{2\alpha}{\tau} k$ **then**
6:          **return** $U_l$
7:      $\lambda_r \leftarrow 2\alpha \|b\|^2$
8:      $\varepsilon \leftarrow \frac{b_{\max}\tau}{2k}$
9:      **while** $\lambda_r - \lambda_l > \varepsilon$ **do**
10:        $\lambda_m \leftarrow \frac{\lambda_l + \lambda_r}{2}$
11:        $U_m \leftarrow$ HISTOGRAMORACLE($b, \lambda_m$)
12:        **if** $\gamma(U_m) > \frac{2\alpha}{\tau} k$ **then**
13:          $\lambda_l \leftarrow \lambda_m$
14:        **else**
15:          $\lambda_r \leftarrow \lambda_m$
16:      $U_l \leftarrow$ HISTOGRAMORACLE($b, \lambda_l$)
17:      $U_r \leftarrow$ HISTOGRAMORACLE($b, \lambda_r$)
18:      $U_l' \leftarrow$ FINDSUBHISTOGRAM($b, U_l, \frac{2\alpha}{\tau} k$)
19:      **if** $\|P_{U_l'} b\|^2 \geq \|P_{U_r} b\|^2$ **then**
20:        **return** $U_l'$
21:      **else**
22:        **return** $U_r$
---

Simplifying and multiplying this iteration bound with the running time of HISTOGRAMORACLE leads to the running time bound stated in the theorem. $\qquad\square$

Theorem 7 now follows directly from Theorem 23. We first restate Theorem 7:

**Theorem 7.** *Let $\zeta > 0$ and $\varepsilon > 0$ be arbitrary. Then there is an $(1 + \varepsilon, \mathbb{U}_k, \mathbb{U}_{c\cdot k})$-approximate tail projection for 2D histograms where $c = O(1/\zeta^2 \varepsilon)$. Moreover, the algorithm runs in time $\widetilde{O}(d^{1+\zeta})$.*

Setting $\xi = O(\varepsilon)$ and $\nu = O(1/\varepsilon)$ gives the $1 + \varepsilon$ guarantee in Theorem 7. Moreover, we use the $\alpha = O(1/\zeta^2)$ dependence from Theorem 3 of [18].

## C.2   Approximate head projection

Next, we show how to construct an approximate head projection from a hierarchical histogram oracle. Similar to the approximate tail projection above, we perform a binary search over the parameter $\lambda$ in order achieve a good trade-off between sparsity and approximation. In contrast to the tail case, we now need an additional subroutine for extracting a "high-density" sub-histogram of a given hierarchical histogram. We reduce this task of extracting a sub-histogram to a problem on trees. Formally, we build on the following lemma about the subroutine FINDSUBTREE.

**Lemma 24.** *Let $T = (V, E)$ be a tree with node weights $w : V \to \mathbb{R}$. Moreover, let $s \leq |V|$ be the target subtree size. Then FINDSUBTREE($T, w, s$) returns a node subset $V' \subseteq V$ such that $V'$ forms a subtree in $T$, its size is at most $2s$, and it contains a proportional fraction of the node weights, i.e., $\sum_{i \in V'} w(i) \geq \frac{s}{|V|} \sum_{i \in V} w(i)$.*

*Proof.* Let $w'$ and $i$ be defined as in FINDSUBTREE. An averaging argument shows that there must be a contiguous subsequence $S$ as defined in FINDSUBTREE with

$$\sum_{j=i}^{i+2s-1} w'(j) \;\geq\; \frac{2s}{2|V|-1} \sum_{j=1}^{2|V|-1} w'(j) \;\geq\; \frac{s}{|V|} \sum_{j \in V} w(j)$$

where the first inequality holds because $S$ contains $2s$ nodes, and the second inequality holds by the construction of the tour $W$.

---
**Algorithm 5** Subroutines for the head projection
---
1: **function** FINDSUBHISTOGRAM($b, U, s$)
2:      Let $T_U = (V_U, E_U)$ be a tree corresponding to the histogram subspace $U$.
3:      Let $w : V_U \to \mathbb{R}$ be the node weight function corresponding to $U$ and $b$.
4:      Let $T_U^*$ be the tree $T_U$ with an additional root node $r$.
5:      Let $w^*$ be defined as $w$ with the root node weight $w^*(r) = \|P_{R_0}b\|^2$.
6:      $V' \leftarrow$ FINDSUBTREE($T_U^*, w^*, s$)
7:      **if** $r \in V'$ **then**
8:           **return** the sub-histogram defined by the splits in $V'$
9:      **else**
10:          Let $r'$ be the root node in the subtree defined by $V'$.
11:          Let $U''$ be a 4-piece hierarchical histogram such that one of the leaf rectangles is $R_{r'}$.
12:          **return** the composition of $U''$ and the sub-histogram defined by $V'$

13: **function** FINDSUBTREE($T, w, s$)
14:      Let $W = (v_1, \ldots, v_{2|V|-1})$ be a tour through the nodes of $T$.          $\triangleright\, T = (V, E)$
15:      Let $w'(j) = \begin{cases} w(v_j) & \text{if position } j \text{ is the first appearance of } v_j \text{ in } W \\ 0 & \text{otherwise} \end{cases}$
16:      Let $S = (v_i, \ldots, v_{i+2s-1})$ be a contiguous subsequence of $W$ with $\sum_{j=i}^{i+2s-1} w'(j) \geq \frac{s}{|V|} \sum_{j=1}^{2|V|-1} w(j)$
17:      **return** the set of nodes in $S$.
---

Let $V'$ be the nodes in $S$. Note that we have defined $w'$ such that every node weight is used only once, and hence we get

$$\sum_{j \in V'} w(j) \;\geq\; \sum_{j=i}^{i+2s-1} w'(j) \;\geq\; \frac{s}{|V|} \sum_{j \in V} w(j)$$

as desired. Finally, since $S$ is contiguous in the tour $W$, the nodes $V'$ form a subtree in $T$ of size at most $2s$. $\hfill\square$

Utilizing Lemma 24, we now show how to extract a "good" sub-histogram from a given hierarchical histogram. More precisely, our goal is to find a sub-histogram $U'$ with a bounded number of histogram pieces that still achieves a comparable "density" $\frac{\|P_{U'}b\|^2}{\gamma(U')} \approx \frac{\|P_U b\|^2}{\gamma(U)}$. In order to precisely state our algorithm and proof, we now formalize the connection between hierarchical histograms and tree graphs.

For a given histogram subspace $U$, let $T_U = (V_U, E_U)$ be the tree defined as follows: First, every split in the hierarchical histogram corresponds to a node in $V_U$. For each split, we then add an edge from the split to the split directly above it in the histogram hierarchy. For a histogram subspace with $\gamma(U)$ pieces, this leads to a tree with $\gamma(U) - 1$ nodes. We also associate each node $v$ in the tree with three rectangles. Specifically, let $R(v)$ be the rectangle split at $v$, and let $R_l(v)$ and $R_r(v)$ be the left and right child rectangles resulting from the split, respectively.

Next, we define the node weight function $w : V_U \to \mathbb{R}$. The idea is that the weight of a node corresponds to the "projection refinement", i.e., the gain in preserved energy when projected onto the finer histogram. More formally, for a rectangle $R$, let $P_R b$ the projection of $b$ onto the rectangle $R$, i.e.,

$$(P_R b)_{i,j} \;=\; \begin{cases} 0 & \text{if } (i,j) \notin R \\ \frac{1}{|R|} \sum_{(u,v) \in \mathbb{R}} b_{u,v} & \text{otherwise} \end{cases}.$$

Then we define the weight of a node $v$ as

$$w(v) \;=\; \left\|P_{R_l(v)}b\right\|^2 + \left\|P_{R_r(v)}b\right\|^2 - \left\|P_{R(v)}\right\|^2 .$$

Let $R_1, \ldots, R_{\gamma(U)}$ be the rectangles in the hierarchical histogram $U$, and let $R_0$ be the $\sqrt{d} \times \sqrt{d}$ "root" rectangle. Since the rectangles are non-overlapping, we have

$$\sum_{i=1}^{\gamma(U)} P_{R_i} b \;=\; P_U b \,.$$

Note that the rectangles $R_1, \ldots, R_{\gamma(U)}$ are exactly the child rectangles of the leaves in the tree $T_U$. Moreover, by the construction of the weight function $w$, we have

$$\|P_{R_0} b\|^2 + \sum_{v \in V_U} w(v) \;=\; \|P_U b\|^2$$

because the contributions from intermediate nodes in the tree $T_U$ cancel out.

**Lemma 25.** *Let $b \in \mathbb{R}^{\sqrt{d} \times \sqrt{d}}$, let $U$ be a hierarchical histogram subspace, and let $s \leq \gamma(U)$ be the target number of histogram pieces. Then* FINDSUBHISTOGRAM$(b, U, s)$ *returns a hierarchical histogram subspace $U'$ such that $\gamma(U') \leq 2s + 4$ and $\|P_{U'} b\|^2 \geq \frac{s}{\gamma(U)}\|P_U b\|^2$. Moreover, the algorithm runs in time $O(d)$.*

*Proof.* Note that by construction, the tree $T_U^*$ defined in FINDSUBHISTOGRAM has $k$ nodes and the node weights $w^*$ satisfy

$$\sum_{v \in V_{T_U^*}} w(v) \;=\; \|P_U b\|^2 \,.$$

Lemma 24 then shows that the subtree defined by the set of nodes $V'$ satisfies $|V'| \leq 2s$ and

$$\sum_{v \in V'} w(v) \;\geq\; \frac{s}{\gamma(U)} \sum_{v \in V_{T_U^*}} w(v) \;\geq\; \frac{s}{\gamma(U)}\|P_U b\|^2 \,.$$

Let $R'_1, \ldots, R'_{|V'|}$ be the leaf rectangles of the subtree $V'$. The above lower bound on the sum of the node weights implies that

$$\sum_{i=1}^{|V'|} \|P_{R'_i} b\|^2 \;\geq\; \frac{s}{\gamma(U)}\|P_U b\|^2 \,.$$

because the rectangles $R'_i$ are non-overlapping and the weights of the inner tree nodes in $V'$ cancel as before. Hence any hierarchical histogram containing the rectangles $R'_1, \ldots, R'_{|V'|}$ satisfies the desired head projection bound. It remains to show that we can convert the subtree defined by $V'$ into a hierarchical histogram.

If the set $V'$ contains the root node of $T_U^*$, the subtree $V'$ directly gives a valid sub-histogram of $U$. On the other hand, if the root node of $T_U^*$ is not in $V'$, we can construct a simple 4-piece hierarchical histogram $U''$ that contains the root rectangle $R_{r'}$ of $V'$ as one of its leaf nodes. The histogram subspace $U''$ is given by four splits corresponding to the boundaries of the root rectangle $R_{r'}$. We can then combine the hierarchical histogram $U''$ with the subtree $V'$ by adding the splits in $V'$ to the hierarchical histogram in $U''$ (by construction, all these splits are valid). The resulting hierarchical histogram then has at most $4 + |V'| \leq 4 + 2s$ pieces.

The running time bound is straightforward: all pre-processing can be accomplished in linear time by computing partial sums for the vector $b$ (projections onto a rectangle can then be computed in constant time). The subroutine FINDSUBTREE also runs in linear time because it requires only a single pass over the tree of size $O(\gamma(U))$. $\qquad\square$

We can now state our approximate head projection algorithm.

**Theorem 26.** *Let $b \in \mathbb{R}^d$, $k \in \mathbb{N}$, and $0 < \tau < 1$. Then* HISTOGRAMHEAD$(b, k, \tau)$ *returns a histogram subspace $U$ such that $\gamma_U \leq \frac{4\alpha}{\tau}k + 4$ and*

$$\|P_U b\|^2 \;\geq\; (1 - \tau) \max_{U' \in \mathcal{H}_{h,k}} \|P_{U'} b\|^2 \,.$$

*Moreover, the algorithm runs in time $O(d^{1+\zeta} \log \frac{\alpha d}{\tau})$.*

*Proof.* First, we introduce a few shortands to simplify notation. Let the histogram subspace $U_l$ be the solution returned by HISTOGRAMORACLE$(b, \lambda_l)$. We then write $h_l = \|P_{U_l} b\|^2$ for the head approximation of $U_l$ and $\gamma_l = \gamma(U_l)$ for the number of histogram pieces in the histogram subspace $U_l$. We adopt a similar convention for $h_r$ and $\gamma_r$ (corresponding to the solution for parameter $\lambda_r$). Finally, let $h^*$ be the optimal head approximation achievable with a $k$-histogram, i.e., $h^* = \max_{U' \in \mathscr{H}_{\mathrm{h},k}} \|P_{U'} b\|^2$.

Rearranging Equation (24), using the new notation, and substituting the optimal $k$-histogram solution for the $\max$-quantifier gives

$$h_l \;\geq\; h^* - \lambda_l \left( k - \frac{\gamma_l}{\alpha} \right) \;. \tag{28}$$

We now consider the case that the algorithm returns in Line 6. We clearly have $\gamma(U_l) \leq \frac{4\alpha}{\tau} k + 4$ when reaching Line 6. Moreover, substituting for $\lambda_l$ in Equation 28 gives

$$
\begin{aligned}
h_l \;\geq\;& h^* - \frac{b_{\max} \tau}{k} \left( k - \frac{\gamma_l}{\alpha} \right) \\
\geq\;& (1 - \tau) h^*
\end{aligned}
$$

where the second line follows from $h^* \geq b_{\max}$. This inequality holds because any histogram with at least 4 pieces can always create a rectangle that isolates the largest element in $b$ (for simplicity, we assume that $k \geq 4$ and $b \neq 0$). Hence $U_l$ satisfies the conditions of the theorem.

Next, we consider the case that the algorithm reaches the binary search. Note that the binary search is initialized and performed such that we have $\lambda_l \leq \lambda_r \leq \lambda_l + \varepsilon$ when it terminates. Moreover, we have $\gamma_r \leq \frac{2\alpha}{\tau} k$ and $\gamma_l > \frac{2\alpha}{\tau} k$. We now distinguish two sub-cases based on the "density" $\frac{h_l}{\gamma_l}$ of the solution $U_l$ corresponding to $\lambda_l$. Let $\phi = \frac{\tau(1 - \tau/2)}{2\alpha}$ be the density threshold compared to the optimal solution density $\frac{h^*}{k}$.

**Sub-case 1:** $\frac{h_l}{\gamma_l} \leq \phi \frac{h^*}{k}$. This inequality allows us to establish an upper bound on $\lambda_l$. Rearranging Equation (28) gives (note that $k - \frac{\lambda_l}{\alpha}$ is negative):

$$
\begin{aligned}
\lambda_l \;\leq\;& \frac{h_l - h^*}{\gamma_l / \alpha - k} \\
\leq\;& \frac{\alpha h_l}{\gamma_l - \alpha k} \;.
\end{aligned}
$$

We now use $\gamma_l \geq \frac{2\alpha}{\tau} k$:

$$
\begin{aligned}
\lambda_l \;\leq\;& \frac{\alpha h_l}{\gamma_l - \tau \gamma_l / 2} \\
\leq\;& \frac{h_l}{\gamma_l} \cdot \frac{\alpha}{1 - \tau/2} \\
\leq\;& \phi \frac{h^*}{k} \frac{\alpha}{1 - \tau/2} \\
\leq\;& \frac{\tau}{2} \cdot \frac{h^*}{k} \;.
\end{aligned}
$$

where we used the density upper bound for $U_l$ valid in this subcase and the definition of $\phi$. Next, we derive a lower bound on $h_r$. Instantiating Equation (28) with $U_r$ instead of $U_l$ gives

$$
\begin{aligned}
h_r \;\geq\;& h^* - \lambda_r \left( k - \frac{\gamma_r}{\alpha} \right) \\
\geq\;& h^* - \lambda_r k \\
\geq\;& h^* - (\lambda_l + \varepsilon) k \\
=\;& h^* - \lambda_l k - \varepsilon k \\
\geq\;& h^* - \frac{\tau}{2} h^* - \frac{\tau}{2} b_{\max} \\
\geq\;& (1 - \tau) h^*
\end{aligned}
$$

where we again used $b_{\max} \leq h^*$. So in this sub-case, $U_r$ satisfies the conditions of the theorem.

**Sub-case 2:** $\frac{h_l}{\gamma_l} \geq \phi \frac{h^*}{k}$. In this subcase, the solution $U_l$ has a good density, so FINDSUBHIS-TOGRAM can extract a good solution with a bounded number of histogram pieces. More formally, since $\gamma_l \geq \frac{2\alpha}{\tau} k$, we can invoke Lemma 25 and get

$$
\begin{aligned}
\left\| P_{U'_l} b \right\|^2 &\geq \frac{\frac{2\alpha}{\tau} k}{\gamma_l} h_l \\
&\geq \frac{2\alpha k}{\tau} \phi \frac{h^*}{k} \\
&\geq \left(1 - \frac{\tau}{2}\right) h^* .
\end{aligned}
$$

Moreover, the output of FINDSUBHISTOGRAM satisfies $\gamma(U'_l) \leq \frac{4\alpha}{\tau} k + 4$, and hence $U'_l$ satisfies the conditions of the theorem.

We can now conclude the proof of the theorem: always, one of sub-case 1 and sub-case 2 holds. Since HISTOGRAMHEAD always returns the best of the two choices $U_r$ and $U'_l$, the overall result has the desired head approximation guarantee.

The overall running time is dominated by the invocations of HISTOGRAMORACLE in the binary search. Each invocation takes $O(n^{1+\zeta})$ time and the number of invocations is the number of iterations of the binary search, i.e., bounded by

$$
\left\lceil \log \frac{\lambda_r^{(0)} - \lambda_l^{(0)}}{\varepsilon} \right\rceil \leq \left\lceil \frac{\lambda_r^{(0)}}{\varepsilon} \right\rceil \leq \left\lceil \log \frac{4\alpha k \|b\|^2}{b_{\max} \tau} \right\rceil .
$$

Since $k \leq d$ and $\frac{\|b\|^2}{b_{\max}} \leq d$, the running time bound in the theorem follows. $\qquad\square$

As before, Theorem 8 follows as a direct consequence of Theorem 26. For completeness, we repeat the statement of Theorem 8:

**Theorem 8.** *Let $\zeta > 0$ and $\varepsilon > 0$ be arbitrary. Then there is an $(1 - \varepsilon, \mathbb{U}_k, \mathbb{U}_{c \cdot k})$-approximate head projection for 2D histograms where $c = O(1/\zeta^2 \varepsilon)$. Moreover, the algorithm runs in time $\widetilde{O}(d^{1+\zeta})$.*

Setting $\tau = O(\varepsilon)$ gives the $1 - \varepsilon$ guarantee in Theorem 8. Moreover, we use the $\alpha = O(1/\zeta^2)$ dependence from Theorem 3 of [18].

### C.3 Recovery of 2D histograms

While we have approximate projections for 2D histograms, they do not suffice to state an overall recovery guarantee in the current form. The issue is that Theorem 9 requires an approximate head projection that is competitive with respect to the sum of subspaces $\mathbb{U} \oplus \mathbb{U}_\mathcal{T}$. While this is easy to satisfy for low-rank matrices (the sum of two rank-$r$ subspace models is contained in the rank-$2r$ subspace model), adding histogram subspace models is more subtle. For instance, consider two $k$-histogram subspaces corresponding to $k$ rows and columns of a $k \times k$ matrix, respectively. The sum of the two subspaces then contains $k^2$ individual rectangles (a chessboard pattern). While these $k^2$ rectangles are not independent (the dimension of the space is only $2k$), the chessboard pattern is no directly contained in the set of $2k$-histogram subspaces. As a result, a head approximation that is competitive with respect to $2k$-histograms is not immediately competitive with respect to the sum of two $k$-histograms.

While head boosting is not directly helpful to overcome this issue, we believe that 2D histograms are "well-behaved" in the sense that boosting is still helpful. In particular, we believe that the sum of two $k$-histograms still allows a constant-factor head approximation with a single $O(k)$-histogram subspace. More formally, we state the following conjecture.

**Conjecture 1.** *Let $c > 0$ be fixed. Then there are universal constants $c_1 > 0$ and $c_2 > 0$ depending on $c$ such that the following holds. For any $b \in \mathbb{R}^d$, there is a $c_1 k$-histogram subspace $U$ such that we have*

$$
\|P_U b\| \geq c_2 \|P_{\oplus^c \mathbb{U}_k} b\| .
$$

If the above conjecture is true, Theorem 26 yields an approximate head projection that is competitive to $\oplus^c \mathbb{U}_k$. Combining this with the boosted version of our recovery framework (see Appendix A.3) then yields an overall recovery algorithm.

# D   Sample complexity of subspace recovery

Here, we establish bounds on the sample complexity of subspace recovery for some particular instances. In particular, our focus is on *fast* sampling operators, i.e., operators that support matrix-vector multiplications with a running time that is *nearly-linear* in the size of the vector. Our results follow from a standard concatenation of previously existing results.

## D.1   Low-rank matrices

Consider the case where the subspace model $\mathbb{U}$ corresponds to the set of rank-$r$ matrices of size $d_1 \times d_1$. Then, the subspace RIP corresponds to the (somewhat) more well-known *rank-r restricted isometry property*, first introduced in Recht, Fazel, and Parillo. We obtain the following result:

**Theorem 27.** *Let $d = d_1^2$. Then, there exists a randomized construction of a matrix $X \in \mathbb{R}^{n \times d}$, with parameters $n = O(rd \operatorname{polylog} d)$, such that $X$ satisfies the rank-r RIP with high probability. Moreover, $X$ supports matrix-vector multiplications with complexity $O(d \log d)$.*

*Proof.* We begin by considering matrices that satisfy the *standard* RIP for $s$-sparse vectors, as well as support fast matrix-vector multiplication. To the best of our knowledge, the sharpest such bounds have been recently obtained by Haviv and Regev (SODA 2016). They show that with high probability, a matrix formed by randomly subsampling $n = O(\delta^{-2} s \log^2(s/\delta) d)$ rows of the discrete Fourier Transform (DFT) matrix satisfies the standard RIP (with isometry constant $\delta$) over the set of $s$-sparse vectors.

Next, we invoke a well-known result by Ward and Krahmer ("New and Improved Johnson-Lindenstrauss Embeddings via the RIP"). Consider a diagonal matrix $D_\xi$, where the diagonal $\xi$ is a Rademacher sequence uniformly distributed over $\{-1, 1\}^d$. Also consider any fixed set of vectors $B$ with $|B| = m$ where $s > O(\log \frac{m}{\eta})$. If $X'$ is any $n \times d$ matrix that satisfies the standard RIP over the set of $s$-sparse vectors with constant $\delta < \varepsilon/4$, then high probability the matrix $X = X'D_\xi$ is a *Johnson-Lindenstrauss* embedding for $E$. Formally, the following is true with probability exceeding $1 - \eta$:

$$(1 - \varepsilon)\|\beta\|_2^2 \leq \|X\beta\|_2^2 \leq (1 + \varepsilon)\|\beta\|_2^2.$$

uniformly for all $\beta \in B$.

Next, we invoke Lemma 3.1 of Candes and Plan ("Tight Oracle Bounds for Matrix Recovery"), who show that the set of vectors corresponding to rank-$k$ matrices, $S_k$, exhibits an $\epsilon$-net $\bar{S}_k$ (with respect to the Euclidean norm) such that

$$|\bar{S}_r| \leq (9/\epsilon)^{(d_1+d_2+1)k}.$$

Also from Candes and Plan, we have that if $X$ is a Johnson-Lindenstrauss embedding with isometry constant $\varepsilon$ for an $\bar{S}_k$, then $X$ satisfies the rank-$k$ RIP with constant $\delta = O(\varepsilon)$. Plugging in $s = O(k(d_1 + d_2))$ and $m = O(s \operatorname{polylog} d)$ and adjusting constants, we get the stated result.

$\square$

## D.2   Histograms

Now, consider the case where the subspace model $\mathbb{U}$ corresponds to the set of (hierarchical or tiling) histograms. Since either type of histogram can be modeled as superpositions of sub-rectangles of the domain $\sqrt{d} \times \sqrt{d}$, we can simply model the histogram subspace model $\mathbb{U}$ as a subset of *dictionary-sparse* vectors $\{x | x = D\alpha, \|\alpha\|_0 \leq k\}$. Here, $D$ is a dictionary of size $d \times \binom{d^2}{2}$ where each column of $D$ corresponds to a single tile (normalized to unit $\ell_2$-norm).

Therefore, any matrix that satisfies the RIP with respect to the dictionary $D$ (abbreviated sometimes as the $D$-RIP) also suffices for reliable histogram subspace recovery. The following result is folklore, and a formal proof can be found in the appendix of Hegde, Indyk, and Schmidt ("Nearly Linear-Time Model-Based Compressive Sensing").

**Theorem 28.** *There exists a randomized construction of a matrix $X \in \mathbb{R}^{n \times d}$, with parameters $n = O(k \log d/k)$, such that with high probability, $X$ satisfies the subspace RIP for the histogram*

Figure 2: (left) Example low-rank matrix of size $d = 133 \times 200, r = 6$. (right) Recovery error of various algorithms as a function of time (2 independent trials).

Figure 3: Running times corresponding to the low-rank matrix recovery experiment in Figure 1. The block Krylov variant of IHT with one iteration has the best running time.

*subspace model. Moreover, X supports matrix-vector multiplications with complexity $O(d \log d + k^2 \operatorname{polylog} d)$.*

# E  Supplemental experiments

We begin with a description of the experimental setup. All experiments were conducted on an iMac desktop computer with an Intel Core i5 CPU (3.2 GHz) and 16 GB RAM. With the exception of the dynamic program (DP) for 2D histograms, all code was written in Matlab. We chose C++ for the 2D histogram DP because it heavily relies on for-loops, which tend to be slow in Matlab. Since the Krylov SVD of [17] is only available as a Matlab routine, we also chose the Matlab version of PROPACK [16] so that the implementations are comparable. Unless reported otherwise, all reported data points were averaged over at least 10 trials.

## E.1  Low-rank matrix recovery experiments

Figure E shows an image of the MIT logo used in the low-rank matrix recovery experiments [15, 19]. For our first experiment, we record $n = 3.5(d_1 + d_2)r = 6994$ linear measurements of the image.

Figure 4: Average approximation errors for the low-rank matrix completion experiment in Figure 1. As for low-rank matrix recovery, the different SVDs achieve essentially the same error.

Figure 5: Results for recovering a hierarchical histogram from subsampled Fourier measurements. As predicted by our theoretical argument, the 2D histogram DP has the best sample complexity.

The measurement operator is constructed by subsampling $m$ rows of a Fourier matrix and multiplying its columns by a randomly chosen Bernoulli vector, similar to the RIP matrix given in Appendix D. The goal is to recover the image from these observations.

We adapt the Singular Value Projection (SVP) algorithm of [15] by replacing the exact SVD step with approximate SVDs (some of which are very coarse), and demonstrate that we can still achieve efficient matrix recovery from few observations. As alternatives to Matlab's in-built svd function, we include the PROPACK [16] numerical linear algebra package, which implements a Lanczos-type method. We also include an implementation of the recent Block-Krylov SVD algorithm of [17], which offers a nice tradeoff between approximation ratio and running time. We test this method with 1 and 8 Krylov subspace iterations (8 is the default provided in the code of [17]).

Figure 3 shows the running times corresponding to the phase transition plot in Figure 1. The only stopping criteria we used were based on a small residual and a maximum number of iterations, so the running times of the algorithms are slowest in the regime where they do not recovery the signal.

The subspace IHT algorithm is iterative, i.e., it produces a sequence of matrix estimates $\{\hat{\theta}^0, \hat{\theta}^1, \ldots, \hat{\theta}^t\}$. Figure E displays the estimation error, $\frac{\|\theta^* - \hat{\theta}^t\|}{\|\theta^*\|}$, as a function of wall-clock time, on two different trial runs. We observe from the plots that PROPACK and the Block Krylov method (with 8 iterations) perform similar to the exact SVD due to the small problem size. Interestingly, a *very coarse* approximate SVD (a single Block Krylov subspace iteration) provides the fastest convergence. Overall, using approximate SVDs within SVP / IHT does not only yield computational speed-ups, but also offers competitive statistical performance.

We also report results of using the SVP / IHT algorithm with approximate projections on a larger matrix completion problem. We generate a matrix of size $d = 2048 \times 2048$ with rank $r = 50$. We only sample $n$ randomly chosen entries of this matrix and attempt to reconstruct the matrix from these entries using SVP with approximate low-rank projections. We vary $n$ and obtain error curves as well as running times. Figure 4 shows the approximation errors for the matrix completion experiment in Figure 1. As for the matrix recovery experiments, all SVDs achieve essentially the same error. We note that the error floor of about 0.05 is a result of our stopping criterion.

### E.2 2D histogram recovery

Finally, we show our results for recovering a 2D histogram from linear observations. As before, we use subsampled Fourier measurements. Our test vector is a $32 \times 32$ hierarchical histogram consisting of 4 rectangles. Hierarchical histograms are essentially 2D piecewise constant functions over a 2D domain where the constant pieces (or tiles) are generated by starting with the entire domain as a single tile and recursively partitioning tiles by making horizontal or vertical splits. We compare three approaches: (i) "Standard" sparsity in the Haar wavelet domain. (ii) Tree sparsity in the Haar wavelet domain [1, 10]. (iii) Our approximate projection algorithm. The focus in our experiments is on sample complexity, so we have implemented only one "level" of the DP in [18]. Figure 5 shows the corresponding phase transitions. The 2D histogram DP does indeed offer the best empirical sample complexity.