[Reviews · NeurIPS 2016]

Reviewer 1

Summary

The authors analyze the recovery problem of vectors fulfilling a general union-of-subspaces model. They discuss approximate versions of projected gradient algorithms for this problem, which can be faster than the projected gradient algorithms using exact projections. Under classical RIP assumptions of the measurement matrix, they prove a recovery guarantee of a general algorithmic framework based on approximate projected gradients if the relevant projections are accurate up to a constant factor. The framework is instantiated for the low-rank matrix recovery problem and the problem of 2D histogram recovery. For the former, this leads to a provable algorithm with the optimal order of sample complexity with faster running time than projected gradient algorithms using exact projections. For the latter, the authors design new, fast approximate projection algorithms. Moreover, the results are complemented by experiments that are in accord with the theoretical findings.

Qualitative Assessment

The paper at hand is well written and accessible. The motivation of the setting is well explained and I consider the topic as very relevant, as the usage of approximate and randomized SVDs in recovery algorithms is ubiquitous in practice, while the implications of their usage as subroutines in algorithms as iterative hard thresholding has not been discussed theoretically in detail yet. In this sense, I consider the paper as an important contribution to the field. As the union-of-subspaces framework is quite general and transcends the two particular cases that are discussed in more detail, I imagine that the contribution of the paper will inspire future work on other specific problems. Despite this positive general quality of the paper, the results about boosted approximate projections are not accurate: The proof of Theorem 16 does not work as detailed in Appendix A.2, as the quantity $c_\mathcal{H}$ should be *squared* in inequality 21 after line 527, to be consistent with the definition of an approximate head projection (Definition 4, lines 121-123). If you continue the following calculations, the resulting approximation constant is worse than the stated bound $2c_{\mathcal{H}}-2c_{\mathcal{H}}^{3/2}+c_{\mathcal{H}}^{2}$. Following the arguments of the authors, we obtain a lower bound of $c_{\mathcal{H}}\sqrt{2c_{\mathcal{H}}+1)}/(1+c_{\mathcal{H}}^2)$, which, on the other hand, is not strong enough for making the proof of Theorem 10 work, as "$f(x)> x$ for $0 < x < 1$" is not true anymore. Therefore, also Theorem 17 cannot be shown as indicated by the authors. This flaw is regrettable, but, in my opinion, affects the overall quality of the paper only partially, as the main results Theorem 9 and Corollary 15 and the instantiation to the low-rank matrix recovery problem do not depend on boosted approximate projections.

Confidence in this Review

2-Confident (read it all; understood it all reasonably well)


Reviewer 2

Summary

The paper is about recovering a vector from linear observations where the vector lies in a union of subspaces. The authors propose a generic algorithm with head and tail approximate projection. Depending on the degree of approximation and running time of the approximate projections, the algorithm can output an accurate estimate. The generic algorithm is instantiated to low-rank matrix recovery and 2d-histogram recovery respectively. The experiments indeed show that the proposed algorithms are fast.

Qualitative Assessment

There is a large amount of similarity between this paper and the journal paper [13]. Some algorithms, theorems, and proofs are replicated almost verbatim. The difference is that [13] focused on problems like compressed sensing with the unknown in a union of subspaces, while the NIPS submission makes a generalization of [13] to encompass low-rank matrix recovery. However, theses are both linear inverse problems with structured sparsity in an atomic norm sense, and the proof structure is almost identical, so this generalization does not seem substantial to me. If the contribution made by this paper is novel and substantial, the authors do not clearly articulate it. Major comments: - Most theorems are written at the abstract level, which makes it a bit hard to follow. - One of the major empirical examples, 2D histograms, has no theoretical support. - For the reason above, I think the real contribution is just Theorem 6. If not, please let us know what the key contributions are. - I could’ve been better if the main content contained the ApproxSVD algorithm or explained it. - Overall, I think the paper is at the borderline. Minor comments: - 162: an => and - Is Krylov SVD the approximate SVD mentioned in Fact 11? - In Figure1, why is 1 iter better than 8 iters?

Confidence in this Review

2-Confident (read it all; understood it all reasonably well)


Reviewer 3

Summary

The paper proposes a general algorithmic framework for provable stable recovery of vectors belonging to a union of subspaces from a noisy low dimensional projection. Based on the notion of approximate tail (resp. head) projection, it provides recovery guarantees for “Approximate Subspace-IHT”, a generic iterative algorithms similar to classical algorithms such as iterative hard thresholding (IHT). The guarantees hold provided that the measurement matrix satisfies a sufficiently strong restricted isometry property on an appropriate (modified) union of subspaces. The main novelties compared to previous work are a) that the framework is valid for a generic union of subspaces (similar to [4], but [4] essentially requires exact projections) b) that it explicitly handles approximate projections (similar to [13], but [13] is restricted to particular unions of subspaces corresponding to some forms of structured sparsity).

Qualitative Assessment

Overall, the contribution is very nicely written, timely, and interesting. Although it seems a natural extension of [13], with proof techniques that are essentially the same, it is very interesting to explicitly express these results which can be of wide applicability and give much freedom to design algorithms with recovery guarantees. Some minor points may deserve some clarification. In Theorems 7 and 8, it is obvious that the constant c = 1/\zeta\epsilon can get large if we want good running time and precision. the discussion of where we pay the price of this large c (apparently in sample complexity) would be welcome here rather than later in the paper. When comparing to [4] there is a discussion on additive approximation error versus relative approximation error. I am not sure that requesting a controlled relative error is a “weaker” assumption than an additive one, especially when the reference error is small. After Theorem 9, it may be worth summarizing the needed assumptions on \eta_0, c_H and \delta for the result to hold. It seems we need 0 < \eta_0 < 1 and \delta/(1-\delta) < c_H < (1+\delta)/(1-\delta). A more explicit expression of the linear convergence implied by the Theorem would also be welcome. page 3: definition of k-wise sum, I assume that for k=1 we get U, not U+U ? page 4: notation \tilde{)} undefined

Confidence in this Review

3-Expert (read the paper in detail, know the area, quite certain of my opinion)


Reviewer 4

Summary

In this paper, the authors consider the problem of signal recovery from a union of subspaces (UoS). UoS problems are a generalization of several structurally constrained problems, such as matrix completion and sparse regression. They show that one can obtain fast algorithms for UoS problems by considering an iterative projection scheme, and obtain highly accurate results even when the projections are approximate. A key insight is that they employ TWO approximate projections, onto what they call the “head” and “tail” subspaces. They apply their results to some canonical cases and show that the proposed methods are faster than methods that require exact projections, while not compromising on accuracy of the recovered signals.

Qualitative Assessment

The paper is well written, and is very relevant. My only issue is that, while the motivation to propose the methods is that of scalability, the authors do not exlpicitly show that their methods are significantly faster than methods requiring exact projections on very large datasets. However, I recommend an accept since I think this addition in the paper is fairly trivial to do. MAJOR COMMENTS: equation (2) only applies if the underlying optimization you intend to solve is least squares with a constraint. In the general case, the second term in the parenthesis will be the gradient of a generic loss function. I expected Definitions 3 and 4 to be symmetric. Is there a reason why c_T = 0 is possible in def. 3 but not in 4? line 160: recent advances in greedy methods obviate the need for the full SVD. Instead, they only require the top singular value to be computed, getting an order of magnitude improvement. Of course, these methods avoid exact projections but it might be prudent to mention the existence of such methods. Line 218: How come there is no dependence on \gamma? won’t a simple counting argument yield \gamma even in a lower bound? Lines 276-278: the explanation here seems to suggest that the order of projections (head first, then tail) is important. Is this correct? What will happen if one reverses the order? Line 287: The U \otimes U_H \otimes U_T RIP condition is different from the usual (U \otimes U \otimes U) RIP conditions one might assume. Can you comment on this distinction and also compare? Maybe provide an example to indicate whether it is tighter or looser. Figures 1-3: I suggest a comparison with other, non SVT based methods for low rank matrix recovery as well. greedily adding one SV at a time is significantly faster, and has been shown to perform very well. Also these greedy projections can be approximate (using power methods for example) A key point you make is that IHT based schemes with approximate projections are faster than SVT based methods. Could you please provide runtimes on larger matrices (order 10K or more? ). More importantly, there should be sizes at which SVT with exact projections becomes infeasible. It might be good to see where that transition occurs. MINOR COMMENTS: Line 63 : is there a reference for results for sparse PCA? Definitions 3 and 4 : Do you need the constants to be bounded by 1 and not 0? In theorem 5, should the \theta in the log terms be \theta^*? line 157: replace “can be” with “is”. you already mention that r << d_1 line 162: an —> and Theorems 7 and 8: what are the typical values of epsilon and zeta that one considers? a small epsilon will mean that zeta has to be large. eps = 1e-2 means that zeta has to be close to 10 atleast right? That makes the run time O(d^11) which is prohibitive. Line 212: Please state the natural structural conjecture here. Line 293: for the sake of completion, please provide the sample complexities for the cases you consider in the main paper as well.

Confidence in this Review

3-Expert (read the paper in detail, know the area, quite certain of my opinion)


Reviewer 5

Summary

This paper considers recovery of a structured vector drawn from an a priori specified union of subspaces, using a fast variant of projected gradient descent. Specifically, the authors show that a method based on so-called approximate projections may simultaneously enjoy analogous error behavior to the "original" methods and a sometimes-significantly reduced computational complexity. Experimental results demonstrate the approach for an example in low-rank matrix recovery.

Qualitative Assessment

The exposition was exceptionally clear, and the ideas presented herein were well-motivated. The essential kernel of the ideas here derive from the approximate-IHT framework proposed in [13] in the context of model-based compressive sensing; the application to low-rank matrix recovery appears to be novel here. That said, the low-rank matrix recovery application examined here appears to be well-matched to the approximate projection framework, since the requisite head and tail approximations may be obtained via an approach by Musco and Musco. The application to histogram recovery also presents as an interesting and viable example within this framework. At first read, the contribution appears to be an application of a known result from [13] to general subspace models, in which case section 3 seems to serve as a lengthy review. Closer inspection suggests that the extension of the approach from structured sparsity models to general subspace models is itself non-trivial, though the authors only describe this nuance in ~2 short paragraphs. It might help place this work in context to make this structural contribution a bit more explicit; it's presented as #1 of 3 contributions, but it really is the main contribution from which the other 2 follow (to my understanding). If this is indeed the case, it might be useful to reorganize the paper slightly, placing the section 3 material before the somewhat lengthy descriptions in sections 2.2 and 2.3.

Confidence in this Review

2-Confident (read it all; understood it all reasonably well)


Reviewer 6

Summary

The paper considers the problem of speeding up algorithms for various problems loosely speaking concerning recovering sparse, structured vectors under RIP measurements, by performing faster approximate projections. The main contribution is to show that even with *constant* factor approximations for the projections (in a suitable sense -- both for the "tail" and for the "head" of the projection) one can get algorithms that are for many cases sample-wise within a constant factor of the optimal sample-complexity, yet are much faster (how much depends on the particular case). The main targets here are low-rank matrix recovery with RIP measurements (actually, where they win most is matrices that are sub-samples of Fourier measurements) and 2D histograms.

Qualitative Assessment

The paper is cleanly written, and studies a well-motivated problem. The main portions of the generic part of the results (Lemma 13 and Theorem 9) are simple and easy to follow, but are very general, and I could imagine being applicable to many other settings on top of what the authors consider. The boosting result (Theorem 16) is also elementary, but is generic enough that it might be easy to improve in specific instances. Overall, I feel like the methods in the paper are accessible to almost everyone, but are very generic, and will be of interest to many in the NIPS community. Some minor/concrete comments: --I would add the boosted version of the algorithm/theorem statement to the paper. (Or at the very least make more explicit that RIP of X is required in the boosted space, rather than the original one.) --Line 525 (apdx): I would use a different letter than r. Somehow, r brings to mind "residual". --Theorem 10 (apdx): I think you should work out the rate for f's convergence to 1, rather than just leaving it as a constant depending on c_H in an unspecified manner. This is actually important, because how much one needs to boost impacts how RIP X needs to be. --Theorem 27 (apdx): The appendix should have a reference section as well. (E.h. line 892, a paper is cited, by writing the author/title. Better to have a proper reference.) --Experiments: An additional good experiment to run would be to compare with non-convex methods with random initialization. (Since the main reason to use this paper in place of those is that the initialization is based on SVD.) --Comparison with [13]: The comparison and overlap [13] needs to be spelled out much better. I'm not particularly familiar with it, but from a quick skim it seems quite similar in terms of tools and proof techniques.

Confidence in this Review

2-Confident (read it all; understood it all reasonably well)